# Seasonality of downward carbon export in the Pacific Southern Ocean revealed by multi-year robotic observations

Léo Lacour[1,2] ✉, Joan Llort [3], Nathan Briggs[4], Peter G. Strutton [1,5] & Philip W. Boyd [1]

At high latitudes, the biological carbon pump, which exports organic matter from the surface ocean to the interior, has been attributed to the gravitational sinking of particulate organic carbon. Conspicuous deficits in ocean carbon budgets challenge this as a sole particle export pathway. Recent model estimates revealed that particle injection pumps have a comparable downward flux of particulate organic carbon to the biological gravitational pump, but with different seasonality. To date, logistical constraints have prevented concomitant and extensive observations of these mechanisms. Here, using year-round robotic observations and recent advances in bio-optical signal analysis, we concurrently investigated the functioning of two particle injection pumps, the mixed layer and eddy subduction pumps, and the gravitational pump in Southern Ocean waters. By comparing three annual cycles in contrasting physical and biogeochemical environments, we show how physical forcing, phytoplankton phenology and particle characteristics influence the magnitude and seasonality of these export pathways, with implications for carbon sequestration efficiency over the annual cycle.

The biological carbon pump (BCP) is considered a major contributor to the Southern Ocean (SO) carbon sink. It removes 2.7 (±0.6) PgC from the euphotic zone (upper ~100 m) annually, representing ~30% of the global BCP[1,2]. Since the BCP concept was initially proposed[3], it has been widely assumed that it was mediated primarily by sinking particulate organic carbon (POC), now termed the Biological Gravitational Pump (BGP)[4,5]. However, present estimates of biological carbon demand in the mesopelagic (~100–1000 m depth) exceed carbon inputs attributed to sinking particles into this stratum by two- to threefold[6,7]. Such discrepancies highlight the need to reassess the pathways that contribute to downward carbon export.

Recently, other particle injection mechanisms (PIPs) have been invoked as additional pathways that help to balance the mesopelagic carbon budget[4]. Studies[8–11] have provided evidence that downward export of organic matter also occurs through localized (1–10 km) eddy-driven subduction of POC. This process, known as the Eddy Subduction Pump (ESP), leads to episodic injection of POC-rich waters below the mixed layer. Similarly, the (sub)seasonal variability of the mixing layer depth leads to detrainment of organic matter to the dark mesopelagic when the mixing layer shoals and conversely entrainment to the surface layer when the mixing layer deepens. This physical mechanism is called the mixed layer pump (MLP)[12–14]. At basin scale, the ESP and MLP export is equivalent to ~25% of the BGP export[4]. However, at the (sub)mesoscale (<100 km) these mechanisms can be equal to, or even greater than the BGP[15,16]. This spatial mismatch illustrates the difficulty of carrying out comprehensive inter-comparison of these pumps in the field. Other PIPs such as the mesopelagic migrant pump[17], seasonal lipid pump[18] and large-scale

[1]Institute for Marine and Antarctic Studies, University of Tasmania, Hobart, Australia. [2]Sorbonne Université, CNRS, Laboratoire d'Océanographie de Villefranche, LOV, Villefranche-sur-Mer, France. [3]Barcelona Supercomputing Center, Earth Sciences Dept., Barcelona, Spain. [4]National Oceanography Centre, Southampton, UK. [5]Australian Research Council Centre of Excellence for Climate Extremes, University of Tasmania, Hobart, Australia. ✉ e-mail: leo.lacour@imev-mer.fr

subduction[19] pump are also important contributors to the BCP, although assessing their relative contribution is even more challenging[4] and beyond this study.

As opposed to the BGP, the physically driven PIPs can potentially transport all types of particles to depth, from small non-sinking particles, including healthy phytoplankton cells, to large fast-sinking aggregates such as fecal pellets[4,8,13]. The composition and size of exported particles will influence their fate in the mesopelagic with respect to depth of remineralisation, sequestration time scale, and consumption by midwater biota[4,15,20]. To determine the relative importance of each of these pathways for carbon sequestration, it is essential to study these mechanisms in the context of the seasonality in characteristics of the upper ocean particle assemblage[4]. Although previous studies[11,21–23] revealed that both the PIPs and BGP vary seasonally, concomitant and extensive in situ observations of these pathways, including upper ocean particle characteristics, are extremely rare. Therefore, little is known about their relative importance over complete annual cycles. Better characterisation of these mechanisms and their seasonality will fundamentally advance our understanding of ocean biological carbon export and sequestration, and help to close regional ocean carbon budgets.

Biogeochemical-Argo (BGC-Argo) floats with multi-year missions and high-frequency sampling offer a promising way to jointly investigate the PIPs and the BGP over a broad range of time and space scales. Such platforms have already been successfully used to characterise the MLP at sub-seasonal to seasonal scales[12,13], the ESP at pan-Antarctic scale[9], and the seasonality of the BGP[22,24]. Here, we jointly investigate these export pathways by refining and linking a range of previously developed techniques. The detection of subsurface anomalous features related to the MLP and ESP[8,12] uses concurrent measurements of chlorophyll $a$ (Chl) fluorescence and particulate backscattering (proxies of phytoplankton concentration and POC), along with oxygen, temperature and salinity (see details in figure captions and Methods). For the BGP, we use two complementary approaches. Spikes in signals of backscattering and chlorophyll $a$ fluorescence provide insights into the sinking of aggregates through the mesopelagic[25]. Analysis of spikes were used in conjunction with short term (order of days) particle accumulation on transmissometer sensors, acting as optical sediment traps (OST; see details in figure captions and Methods)[26]. This combination of approaches links a bespoke sensor constellation with recent advances in bio-optical signal analysis to compare the BGP with two PIPs, the MLP and the ESP, through three annual cycles. We focus on a single float, which sampled contrasting biogeochemical provinces across the Pacific sector of the SO, to infer the causal mechanisms that set the magnitude and seasonality of each of these pathways.

## Sampling across Southern Ocean biogeochemical provinces

We present in situ observations collected by a BGC-Argo float (WMO 7900791) that travelled >6000 km across the Pacific sector of the SO (Fig. 1a). This float was deployed in May 2016 in sub-polar waters and sampled the water column from 500 m to the surface every 1–5 days during three annual cycles. It moved south across the Polar Front in September 2016 as evidenced by the abrupt decrease in surface temperature (from 6 °C to 3 °C, Fig. 1c). The float remained in the vicinity of the Polar Front, a region of intense eddy kinetic energy (Fig. S12), for almost an annual cycle, until July 2017. Then, it entered polar waters where it measured near-zero surface temperatures. From June to August 2018, the float was close to the sea-ice edge (Fig. 1b) in an area characterized by very low salinity, and highly stratified surface waters (Fig. 1d and S1c) which prevented any deep mixing events during winter. Indeed, the mixing layer depth, defined here as the maximum vertical gradient in Chl ($MLD_{bio}$, see Methods), remained shallow (<100 m) as opposed to the two previous winters where mixing reached 250–350 m (the black thick line in Fig. 1c, d). The float ended its mission in May 2019. This long trajectory across different oceanic

provinces enabled the characterization of the export pathways over a broad range of time and space scales in contrasting environments. The downside of this long sampling trajectory is the difficulty in studying the seasonality of these pathways when the float moves across different water masses. The trajectory was therefore segmented into three bloom cycles during which the contiguous nature of water masses was verified (Fig. S2 and Methods).

## Characteristics of the particle assemblage

Environmental forcing and ocean physics partly shape the seasonality of phytoplankton stocks and community composition, hereafter called phenology, and by extension the characteristics of the particle assemblage in the upper ocean[27]. These characteristics will ultimately influence how particles are transported into the oceans' interior via gravitational settling or physical injection[5,20]. We thus commence by discussing the differences in particle characteristics across the three annual cycles before analysing, in the next section, their ramifications for the seasonality of the BGP, the ESP and the MLP.

The contrasting environments between the three annual cycles corresponded to clear differences in bio-optical proxies for concentration, size and composition of phytoplankton community and associated particles (Fig. 2). The 2016–17 bloom, near the Polar Front, was the most intense as reflected in POC, derived from the particulate backscattering coefficient $b_{bp}$ (up to 175 mg C m$^{-3}$, Fig. 2a and Methods), and the concentration of small (<100 μm) fluorescing particles $Chl_s$ (Fig. S9 and Methods). This bloom was characterised by very low $Chl_s/b_{bs}$ values (where $b_{bs}$ is the backscattering coefficient of <100 μm particles), a ratio which varies according to changes in phytoplankton community structure, photoacclimation and nutrient status[28–30]. Here, we attribute low $Chl_s/b_{bs}$ values to the growth of coccolithophores, calcifying phytoplankton which form liths, bio-mineral shells with a high refractive index and thus high backscattering signal[31]. Coccolithophore blooms are recurrent features in the SO, conspicuous from space[32,33]. Indeed, satellite records confirmed the presence of an intense coccolithophore bloom during this summer in the area where the float was profiling (Fig. S3). Bio-minerals act as ballast by increasing particle specific gravity and sinking speeds, and protecting POC from remineralisation[34], enhancing the transfer efficiency of particles trough the mesopelagic. This likely explains the massive invasion of POC to 450 m following the coccolithophore bloom (Fig. 2a). Such particle characteristics should typically increase the relative importance of the BGP compared to physically driven pathways[35]. Our POC estimates might be overestimated here, due to high backscattering by these bio-minerals.

The 2017–18 bloom was less intense with lower POC levels (~50 mg C m$^{-3}$, Fig. 2a), but high $Chl_s/b_{bs}$ values at its peak (apex, Fig. 2a, b). Such surface high $Chl_s/b_{bs}$ ratios associated with high-latitude spring blooms have been attributed to diatom-dominated events[28,29], which in this case would have been stimulated by dissolved iron supply from winter deep mixing and sea-ice melt (Fig. 1d). An iron stress index ($\alpha_{NPQ}$) was derived from the non-photochemical quenching signal in float fluorescence profiles[36] (see Methods). Higher $\alpha_{NPQ}$ values in Fig. 2c correspond to greater phytoplankton community iron stress. The decrease in POC (Fig. 2a) and mean size of particles (Fig. 2c and Methods), in the surface layer after the bloom apex in December 2017 (Fig. 2a, c), coincided with an increase in phytoplankton iron stress (Fig. 2c, grey bars), suggesting that the diatom bloom terminated due to iron limitation. Iron status is a key factor that influences particle characteristics. Indeed, the evolution from iron-replete to iron-limited phytoplankton as the bloom progresses is generally correlated with a change in the buoyancy of particles produced in the surface layer, from neutrally buoyant particles to fast-sinking agregates[37].

The 2018–19 bloom was only partially sampled as data were missing for nearly two months early in the productive season, likely due to the presence of sea-ice which prevented any data transmission.

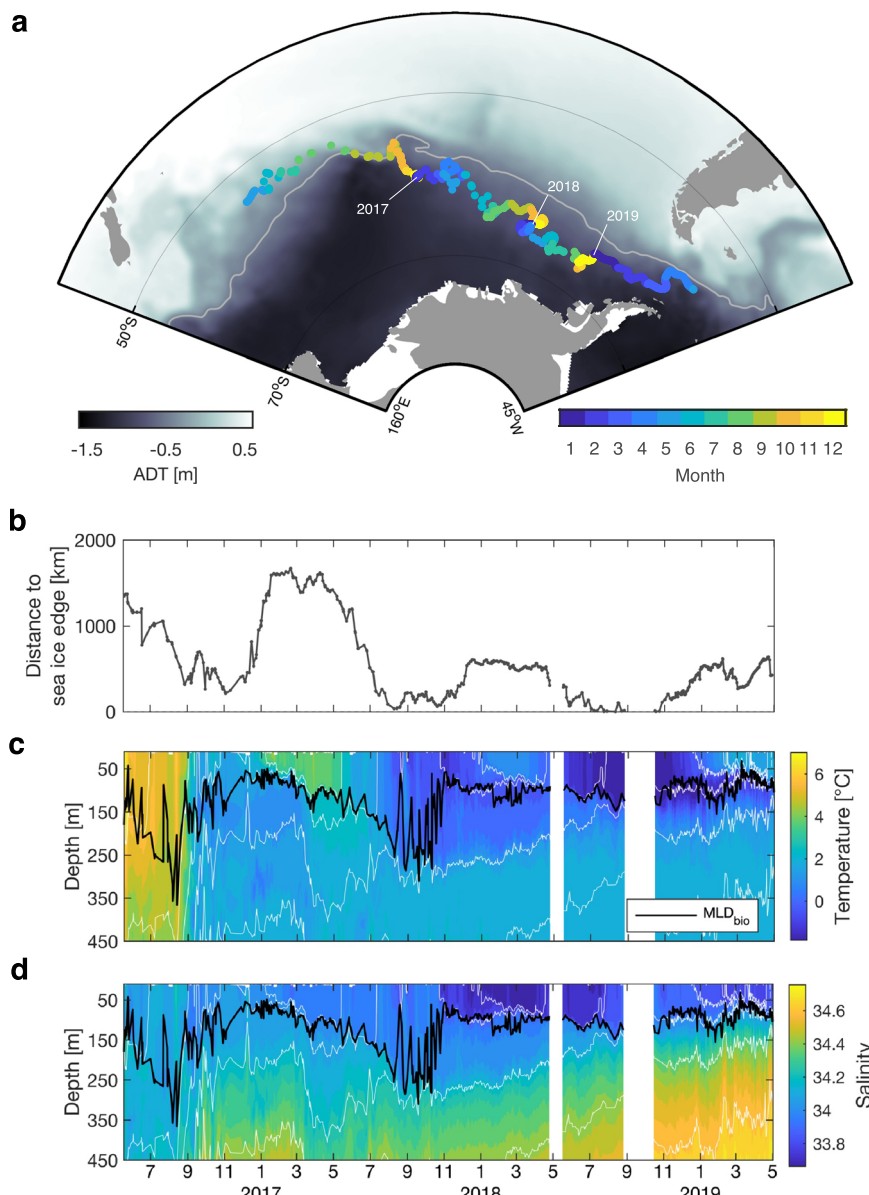

**Fig. 1 | Contrasting environmental conditions in the Pacific sector of the Southern Ocean. a** Float surfacing positions (every 1–5 days) during its 36-month mission. Background map is a climatology of Absolute Dynamic Topography (ADT). Light grey line indicates the climatological position of the Polar Front (ADT = −0.48 m) used in Ardyna et al.[72] and derived from Swart et al.[73]. **b** Minimum distance between float positions and the sea ice edge, defined as the 15% sea ice concentration limit. Vertical sections of **c** temperature and **d** salinity recorded by the float. Light grey lines are isopycnals and the dark line is the mixing layer depth as defined by biological criteria ($MLD_{bio}$, see Methods). Periods with missing data are blank in panels **c** and **d**.

In January/February 2019, the presence of a subsurface chlorophyll maximum, characterized by high $Chl_s/b_{bs}$ values at a depth of $MLD_{bio}$ due to photoacclimation (Fig. 2b), relatively small particles in the surface layer (Fig. 2c), and high iron limitation index (Fig. 2c) suggest that the bloom was at a late stage when sampled, typical of open-water conditions in the seasonal sea-ice zone[38]. The second intense peak of POC recorded in April 2019 was observed in the Drake Passage, near the Antarctic Peninsula, a region known to be influenced by iron-enriched shelf waters[39]. Accordingly, the iron limitation index dropped to low values similar to those observed during the first 2016–17 bloom (Fig. 2c).

### Seasonality of multiple carbon export pathways
Here, we explore the effect of phytoplankton phenology and seasonality in particle characteristics on the seasonality of the BGP, ESP and MLP.

The BGP was characterised by two independent methods, the optical spike and the optical sediment trap approaches. Optical spikes were used to quantify the concentration of large (>100 μm) fast-sinking fluorescing ($Chl_l$) and backscattering ($b_{bl}$) particles in the water column (Fig. S5), from which downward fluxes $F_{Chl\ spike}$ and $F_{POC\ spike}$ were derived by multiplying these concentrations with typical meso-pelagic particle sinking speeds in high latitude blooms (see Methods). A constant bulk sinking speed is assumed here because information on time-varying sinking speeds was not available. However, uncertainty in bulk sinking speed derived from plume tracking was propagated to uncertainty in our $F_{POC\ spike}$ estimates (see Methods). During the 2016–17 bloom, both $F_{Chl\ spike}$ and $F_{POC\ spike}$ peaked prior to the maximum in depth-integrated POC (i.e., stock) over the mixing layer (early January, Fig. 3a) that was attributed to the coccolithophore bloom. $F_{POC\ spike}$ reached a maximum value of ~900 mg C m$^{-2}$ d$^{-1}$ in early December when the mean particle size and $Chl_s$ stock in the upper

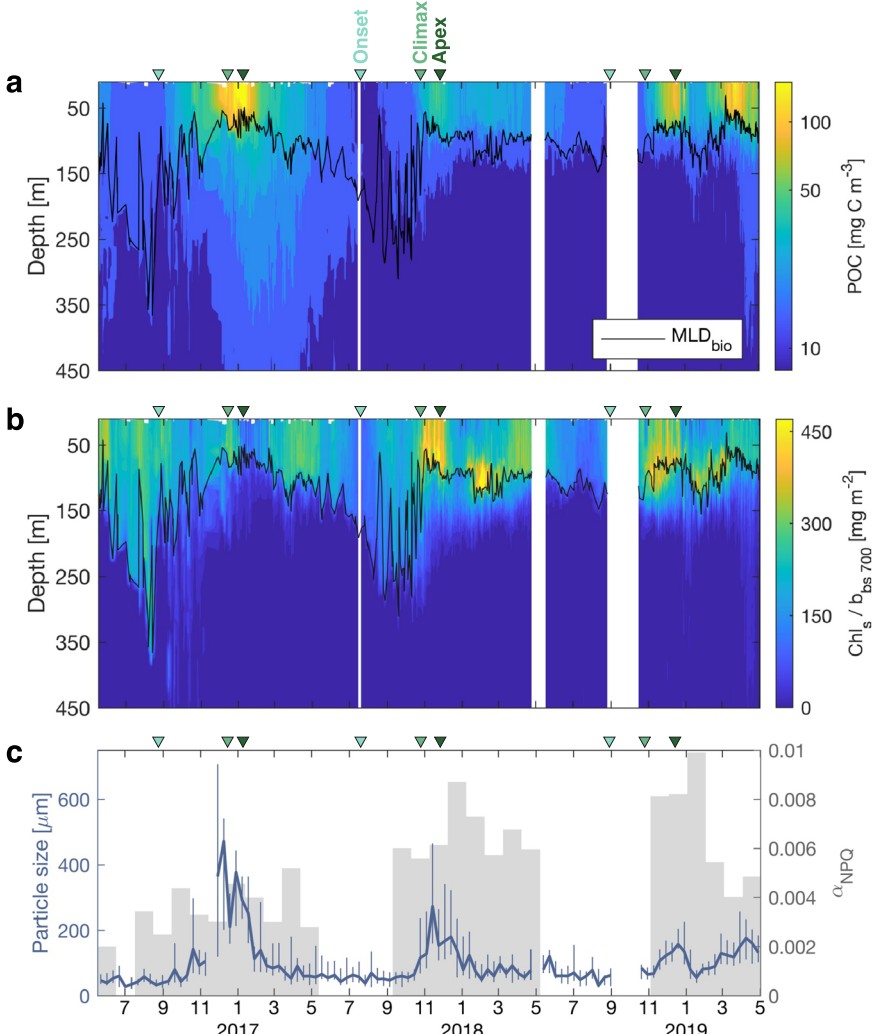

**Fig. 2 | Characteristics of the particle assemblage over three annual cycles, quantified by float-derived bio-optical proxies.** Timeseries of **a** particulate organic carbon (POC) derived from $b_{bp}$, and **b** chlorophyll to backscattering ratio $Chl_s / b_{bs}$. Black lines show the mixing layer depth ($MLD_{bio}$). **c** Mean particle size (diameter) in the upper 50 m estimated from high-frequency variations in $b_{bp}$ and the beam attenuation coefficient $c_p$ using the variance-to-mean ratio method[52] (see Methods). The blue line represents the median and vertical lines represent inter-quartile ranges, over 10-d bins. Grey bars show the iron stress index $\alpha_{NPQ}$ averaged over 30-d bins. $\alpha_{NPQ}$ was derived from the non-photochemical signal in float fluorescence profiles (see Methods). As milestones, triangles on top of each panel mark the timing of onset (first positive net phytoplankton accumulation), climax (maximum accumulation) and apex (maximum phytoplankton carbon stock) for each bloom, following the method in Uchida et al. 2019[68] (see Methods).

layer were also maximum (Figs. 2c and 3a). The high $F_{Chl\ spike}/F_{POC\ spike}$ ratio in the mesopelagic (Fig. S7) suggests that the exported material was dominated by fresh labile phytoplankton aggregates. Inversely, low $F_{Chl\ spike}/F_{POC\ spike}$ following the coccolithophore bloom indicates an export of detrital matter with potentially high mineral content such as coccoliths associated with fecal pellets. Therefore, the seasonal succession in dominant phytoplankton groups, reflected in the different timing of $Chl_s$ and POC peaks, influences the lability of aggregates in the mesopelagic and potentially their penetration depth (Fig. S6). Despite the lowest surface POC and $Chl_s$ stocks of all three blooms, the 2017–18 bloom had relatively high $F_{POC\ spike}$ (~300 mg C m$^{-2}$ d$^{-1}$, similar to the 2018–19 bloom) and $F_{Chl\ spike}$ values, and the highest $F_{Chl\ spike}/F_{POC\ spike}$ ratio. Iron limitation during the 2017–18 diatom bloom may have led to high level of aggregation as indicated by the high mean particle size in the surface layer (Fig. 2c), also reported in another study in the subarctic Pacific[37].

The OST flux measured at 300 m was divided into two components, a continuous flux of small slow-sinking particles and a pulsed flux of large fast-sinking particles (see Methods)[26]. The 2016–17 bloom illustrates the decoupling between the continuous and pulsed fluxes

(Fig. 3c). The latter reached ~1100 mg C m$^{-2}$ d$^{-1}$ in mid-January following the coccolithophore bloom, while the continuous flux peaked 20–30 days later in early February 2017 and was an order of magnitude smaller. The delay between the continuous and pulsed flux can be explained by the differential sinking speed between small and large particles, but may also reflect disaggregation processes in the meso-pelagic where large particles from the pulsed flux fragment into smaller particles later detected in the continuous flux record[25]. Another important feature revealed by the OST is the relatively high continuous flux in late summer and fall 2018 when mixing layer POC (Fig. 3a) and particle size (Fig. 2c) were close to the annual minimum. This late long-lasting flux of slow-sinking small particles contributed ~60% of the annual continuous flux and was equivalent to ~25% of the total OST flux (continuous + pulsed; Fig. S13).

The eddy subduction pump contribution to the BCP was quantified by first identifying subducted water parcels along the float trajectory. Subducted water parcels were detected as subsurface anomalies in POC, spice (density-compensated changes in temperature and salinity) and oxygen profiles (Fig. S11 and Methods). Such features were detected only during 2016–17 in the vicinity of the Polar

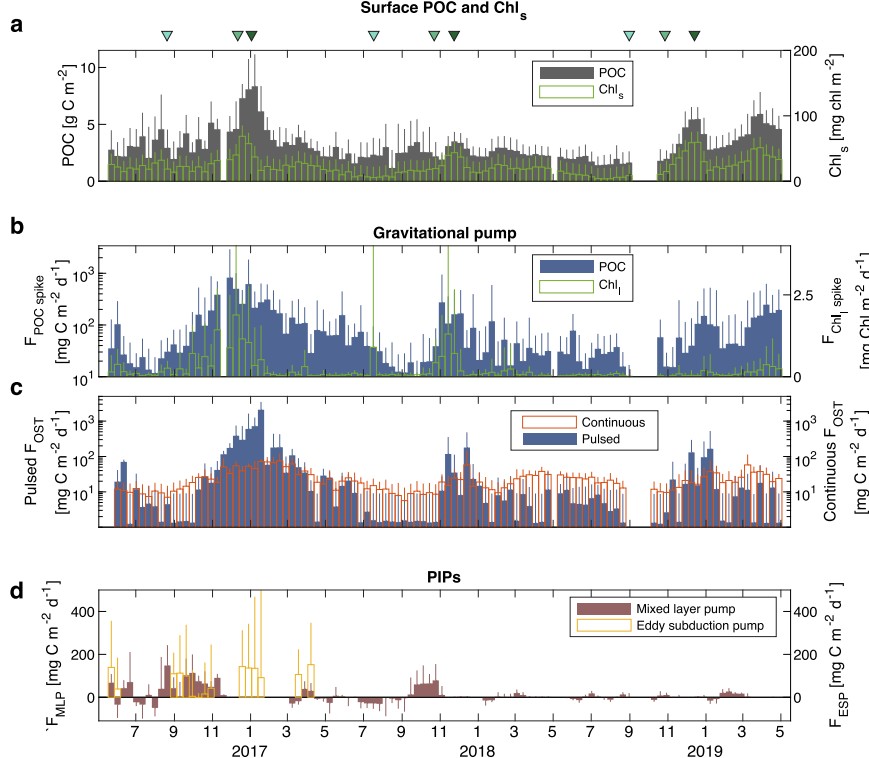

**Fig. 3 | Seasonality of multiple carbon export pathways. a** Stock of particulate organic carbon (POC) and small-particle chlorophyll (Chl$_s$) in MLD$_{bio}$. **b** Flux of large-particle POC (blue bars) and chlorophyll (green bars) in a 100-m bin below MLD$_{bio}$, derived from b$_{bl}$ and Chl$_l$ respectively (see Methods). **c** Continuous flux of small slow-sinking particles (slowly accumulating on the transmissometer window; red bars) and pulsed flux of large fast-sinking particles (creating discontinuities in transmissometer records; blue bars) measured by the optical sediment trap (OST) at ~300 m (see Methods). Panels **b** and **c** show two complementary approaches to

characterise the BGP. **d** POC export driven by two particle injection pumps (PIPs): the mixed layer pump (MLP, brown bars) and the eddy subduction pump (ESP, yellow bars). In all panels, bars represent a 10-d median of the stock or flux. The length of the error bars represents the interquartile range (see Methods). Similarly to Fig. 2, triangles on top of panel **a** mark the timing of onset (first positive net phytoplankton accumulation), climax (maximum accumulation) and apex (maximum phytoplankton carbon stock) for each bloom, following the method in Uchida et al.[68] (see Methods).

Front, a highly energetic region prone to (sub)mesoscale circulation and ESP events[6] (Fig. S12). No clear seasonal patterns were observed in ESP fluxes (F$_{ESP}$), with a maximum export of ~180 mg C m$^{-2}$ d$^{-1}$ on average in 10-d bins in June, September and December 2016, January and April 2017 (Fig. 3d). This temporal distribution suggests that a great diversity of particle types can be exported trough this pathway. Indeed, ESP events in September occurred early in the productive season prior to the bloom climax (Fig. S13). Thus, these events actively transported small and freshly produced organic material as deep as ~400 m (Figs. 2c, 3d and S13). Such labile particles would likely not have reached these depths via the BGP, which illustrates the biogeochemical significance of the ESP. A second train of ESP events occurred concurrently with the main peak in surface Chl$_s$ stock and mean particle size (December–January 2016) suggesting that the ESP also transported large fresh aggregates produced in the euphotic zone, in addition to small suspended particles and dissolved compounds[13]. We cannot rule out the hypothesis that subsurface anomalous water parcels sampled by the float were potentially subducted weeks/months earlier and persisted at depth, as suggested in a study in the North Atlantic[21]. However, the strict detection thresholds that we applied for oxygen, spice and POC anomalies should select only recent ESP events.

The mixed layer pump flux (F$_{MLP}$) was quantified as the sum of the POC stock detrained from and entrained into the productive layer due to changes in the mixing layer depth during a given time period (see Methods)[13,14]. The mean flux over 10-d bins is reported in Fig. 3d (brown bars). Negative values indicate that the entrained POC was greater than the detrained POC during the 10-d period.

Conversely, positive values indicate a net POC export out of the productive layer. POC export occurred during the transition between winter deep mixing and spring stratification (2-3 months), as also reported in previous work[12,16], with maximum values in late August 2016 (~140 mg C m$^{-2}$ d$^{-1}$). Lower seasonal variation in the mixing layer depth in 2018–19 due to the influence of melt water resulted in lower F$_{MLP}$ values (Fig. 3d). Time-integration of F$_{MLP}$ over complete annual cycles demonstrates that the MLP is an efficient pathway to transfer POC below the maximum winter mixing layer depth (Fig. S13). Indeed. the net export of POC was 5.7 g C m$^{-2}$ in 2016–17 and 2.3 g C m$^{-2}$ in 2017–18 (not calculated in 2018–19 due to missing data for nearly two months). The strong sub-seasonal variability of the mixing layer depth, with intermittent stratification and deep mixing events, also led to transient inputs of fresh organic material as deep as 350 m, as revealed by high Chl$_s$/b$_{bs}$ ratio in the mesopelagic zone (Fig. S8). These transient inputs may sustain the mesopelagic biota before the main export pulse following the spring bloom[12,13]. The last export events, in early November 2016 and late October 2017, before the permanent summer stratification occurred at a time when particles in the surface started to increase in size (Fig. 2c), suggesting that relatively large particles can also be actively transferred to depth, similar to the ESP. The clear difference in the depth strata of particles injected by the two PIPs (Figs. S11 and S14) indicate that POC stocks exported by these physically mediated pathways were not double counted. However, these results illustrate the possible interplay between a PIP and the BGP, where large particles could sink further following a physical injection event.

## Implications for oceanic carbon storage

Multi-year and high-resolution float observations provide a new and more comprehensive view of export pathways that contribute to the BCP over the annual cycle. The main insight from our study is the seasonal succession of mechanisms contributing to a long and sustained transfer of organic carbon to the deep ocean. Export due to the ESP was observed early in the bloom, before the bloom climax, and again at the bloom apex, but only in one of three years. The MLP was most active during the seasonal transition from deep mixing to stratification (between bloom onset and climax), and again during subseasonal stratification events. There was large interannual variability in the magnitude of the MLP. Finally, the gravitational pulsed flux was most intense at the bloom apex, while the continuous flux persisted throughout the year, with maximum values delayed in time with respect to the bloom apex—in fall in the case of the 2017–18 bloom. The contribution of all these processes over a complete annual cycle should be accounted for when computing regional mesopelagic carbon budgets.

Total annual POC export from small slow-sinking particles (continuous OST flux) reached 11 g C m$^{-2}$ in 2016–17 and 8 g C m$^{-2}$ in 2017–18 (Fig. S13), of which 43 and 27% was exported prior to the bloom apex during the first and second annual cycles, respectively. As for large fast-sinking particles (pulsed OST flux), these relative contributions increase to 65 and 52%, with total annual export of 53 g C m$^{-2}$ and 5 g C m$^{-2}$ in 2016–17 and 2017–18, respectively (similar values for the optical spike approach). These results stress the importance of persistent observations of the BGP, not only following spring bloom collapse. In comparison, total annual export from the MLP was 5.7 g C m$^{-2}$ in 2016–17 and 2.3 g C m$^{-2}$ in 2017–18 representing 9 and 17% of total export from the BGP (pulsed + continuous OST flux). This contribution falls within the range estimated in the North Atlantic with a similar approach (5–25%)[14], and is slightly lower than the average contribution at high latitudes (23%)[13]. These relative contributions increase to 50 and 165% at the time of the bloom climax in 2016–17 and 2017–18, respectively (Fig. S13), indicating that the MLP is the dominant carbon export pathway during the pre-bloom period when strong variability in the mixing layer depth occurs. Evaluating the total annual export from the ESP is more challenging due to the difficulty to integrate fluxes from episodic and local ESP events[23]. Our method also does not estimate net ESP export as upward fluxes are not considered[16]. We thus compare the mean gross ESP export with total BGP export solely at the time of the bloom apex. In 2016–17, the ESP contribution to total BGP export was about 19%, a value similar to the North Atlantic spring bloom (~25%)[8] and at the peak bloom in the Drake Passage (~18%)[23]. The absence of ESP events during the second and third annual cycles in less energetic regions (Fig. S12) suggests that, in agreement with Llort et al.[9], this pathway might be less widespread in the Southern Ocean than previously reported in the model-based study of Omand et al.[8]. On the other hand, episodic and local ESP events might not be efficiently captured by a single BGC-Argo float, in contrast to a glider, which can sample regions of interest such as fronts at high resolution. However, the emerging global float network could overcome this limitation and provide a valuable statistical view of the importance of the ESP at the basin scale.

A final important insight from our study is the identification of the timing and depth strata for the potential interplay between the PIPs and BGP[4]. Float-derived optical proxies revealed that PIPs potentially inject large fresh aggregates as deep as 400 m, in addition to suspended particles and dissolved organic matter. These fresh aggregates can then sink further through gravitational sinking[35]. With vertical velocities of order of 100–1000 m d$^{-1}$[40,41], physically mediated processes, such as the MLP and ESP, considerably accelerate the transit of particles through the main remineralisation horizon. The interplay between the PIPs and BGP makes it challenging to assess their relative contribution to carbon sequestration, but the overlap that we find in their timing clearly suggest that their joint contributions can boost the overall efficiency of the BCP.

## Methods

### Float data processing

The BGC-Argo float used is a Teledyne Webb Research APEX float, equipped with a Sea-Bird SBE41 CTD sensor, ECO fluorometer and scattering sensors measuring chlorophyll *a* fluorescence and the volume scattering function (at -124°, 470 and 700 nm wavelengths), OC4 radiometer measuring downwelling irradiance at 412, 443, 490 and 550 nm, C-Rover transmissometer measuring the beam attenuation coefficient at 660 nm ($c_p$, in m$^{-1}$), and an Aanderaa Oxygen Optode.

The float mission included CTD and bio-optical day and night profiles every 1.5–6 days, from 500 m depth to the surface, with a parking depth of 300 m. The vertical sampling resolution ranged from 3 to 10 m depending on the float ascent speed. All profile data were interpolated at 1 m resolution.

The CTD and trajectory data were quality-controlled using the standard Argo protocol[42]. ECO raw signals were converted to chlorophyll *a* concentration (Chl, in mg Chl m$^{-3}$) and particulate backscattering coefficient ($b_{bp}$, in m$^{-1}$) following the BGC-Argo procedures[43,44]. Bio-optical data were quality-controlled following the BGC-Argo quality control manual[45]. In addition, Chl was corrected for non-photochemical quenching (NPQ) following Xing et al.[46]. Briefly, for each profile where the sun elevation angle was >5°, the maximum Chl value above the mixed layer depth, defined here as a density difference of 0.01 kg m$^{-3}$ with a reference value at 5 m depth, was extrapolated toward the surface. As an additional condition, the depth of the extrapolated Chl value had to be shallower than the 15 µmol photons m$^{-2}$ s$^{-1}$ isolume[46]. As the instantaneous Photosynthetically Available Radiation (iPAR) was not directly measured, we estimated iPAR profiles from the measured downwelling irradiance at 4 wavelengths. At each depth, a spline interpolation was used to compute the irradiance spectra from 400 to 1000 nm at 1 nm resolution, with irradiance value set to zero at 1000 nm. iPAR was then calculated by integrating the interpolated irradiance spectra from 400 to 700 nm. Oxygen data were calibrated using air measurements following Johnson et al.[47], with a mean gain factor of 1.0557. The $b_{bp}$ at 700 nm was converted into particulate organic carbon (POC, in mg C m$^{-3}$) using the SO-specific relationship in Johnson et al.[48]. POC = 3.12*10$^4$($\pm$2.47*10$^3$)*$b_{bp700}$ + 3.04($\pm$6.78). Similarly, the $b_{bp}$ at 470 nm was converted into phytoplankton carbon ($C_{phyto}$, in mg C m$^{-3}$) following Graff et al.[49]: $C_{phyto}$ = 12.128*10$^3$*$b_{bp470}$ + 0.59. Note that these carbon values might be over-estimated during the 2016–17 coccolithophore bloom due to the high refractive index of coccoliths.

In this study, we distinguish between the mixed layer, the zone of relatively homogeneous water formed by the history of mixing, and the mixing layer, the zone in which mixing is currently active. The mixing layer depth was computed as the maximum vertical gradient of Chl (MLD$_{bio}$) following Lacour et al.[12]. The underlying concept is that Chl, as a proxy of phytoplankton concentration, is homogeneous over the whole mixing layer if turbulent mixing overcomes vertical variations in phytoplankton net growth rate[50]. MLD$_{bio}$ should capture the high-frequency variability of the mixing layer at time scales typical of phytoplankton growth[51]. Note that in summer stratified waters, phytoplankton can thrive below the mixing layer. In this situation, MLD$_{bio}$ corresponds more likely to the base of the euphotic zone.

### Mean particle size

Mean particle diameter in the upper 50 m, weighted by particle cross-sectional area, was estimated from high-frequency variations in $b_{bp}$ and $c_p$ using the variance-to-mean ratio method[52], adapted for use on profiling floats following Rembauville et al.[29]. This method extracts particle size information from the "spikiness" of optical profiles where

particle concentrations are too high to separate individual spikes from a small-particle baseline. Briefly, mean diameter was estimated from $c_p$ via Eq. (1):

$$\bar{A}_{cp} = \frac{\text{var}(c_{pdetrended})}{mean(c_p)} \frac{V}{Q_c} \frac{1}{\alpha(\tau)} \tag{1}$$

$$\alpha(\tau) = \begin{cases} 1 - (3\tau)^{-1}, & \text{if } \tau \geq 1 \\ \tau - \frac{\tau^2}{3}, & \text{if } \tau \leq 1 \end{cases}$$

$$\tau = \left( \frac{t_{res}}{t_{samp}} \right)$$

$$\bar{D}_{cp} = 2\sqrt{\bar{A}_{cp}\pi^{-1}}$$

where $t_{res}$ is the residence time of a particle in the sample volume (0.1 s), $t_{samp}$ is the duration of a single measurement (1 s), $V$ is the transmissometer sample volume (12.5 ml), $Q_c$ is the optical attenuation efficiency of the particles (assumed to be 2 following Bohren and Huffman[53]), $\text{var}(c_{p\ detrended})$ is the variance of $c_p$ after detrending using an 5-point running median, and $D_{cp}$ is the area-weighted mean particle diameter. The calculation was then repeated for $b_{bp}$ at both 470 and 700 nm, by replacing $c_p$ with $b_{bp}$ in Eq. 1, and using $V = 0.62$ ml (Briggs et al.[52]), $t_{samp} = 1$ s, $t_{res} = 0.06$ s, and $Q_{bb} = 0.02$[52]. In order to reduce noise in the individual size estimates, the three estimates were then combined together into 10 d bins, whose medians and 25th and 75th percentiles are reported here.

## Optical spikes in the mesopelagic

Chl and $b_{bp}$ were partitioned into six components following Briggs et al.[54]: deep sensor blanks, including a background of small refractory particles ($Chl_r$ and $b_{br}$); small, labile fluorescing ($Chl_s$) and backscattering ($b_{bs}$) particles; and large, fast-sinking fluorescing ($Chl_l$) and backscattering ($b_{bl}$) particles. The division between small and large corresponds approximately to a particle chlorophyll content of 60 pg for $Chl_s$ versus $Chl_l$ and a particle diameter of 100 μm for $b_{bs}$ versus $b_{bl}$. Timeseries of $Chl_l$ and $b_{bl}$ at 470 and 700 nm are shown in Supplementary Fig. S5. $Chl_l$ corresponds to fresh phytoplankton aggregates while $b_{bl}$ additionally includes fecal and detrital matter[25]. The vertical fluxes of large particles were computed as follows:

$$F_{Chl\ spike} = \langle Chl_l \rangle * w_{Chl_l}$$

$$F_{POC\ spike} = \langle b_{bl700} \rangle * Q_{b_{bl}/POC} * w_{b_{bl}}$$

where $w_{b_{bl}} = 74(\pm 45)\,md^{-1}$ and $w_{Chl_l} = 98(\pm 64)\,md^{-1}$ are the mean (and standard deviation) sinking speeds of large backscattering and fluorescing particles, respectively, as estimated in Briggs et al.[54]. These estimates are based on 18 particle export pulse events for $b_{bl}$ and 25 events for $Chl_l$, including 22 events in total in the Southern Ocean. $Q_{b_{bl}/POC} = 3.12*10^4(\pm 2.47*10^3)$ is the slope of the relationship between $b_{bp}$ and POC developed in Johnson et al.[48]. Given that $b_{bl}$ is a fraction of $b_{bp}$, the intercept of the relationship was not used. Brackets represent a 10-d bin average of all $Chl_l$ and $b_{bl}$ values in a 100-m bin below the mixing layer depth. The sinking speed w is assumed to be constant over time. The probability distribution of the flux F was derived from the mean and standard deviation of each term in the calculation, using a Monte Carlo simulation, with 5000 iterations, and assuming each term is normally distributed. The mean and standard deviation of $\langle Chl_l \rangle$ and $\langle b_{bl700} \rangle$ represent the uncertainty introduced by the spatio-temporal variability. Vertical bars in Fig. 3b show the median of the

distribution of F while the length of error bars represent its interquartile range. Similarly, Fig. 3a shows the median and interquartile range of POC and Chl stocks in the mixing layer, with uncertainties introduced by the $b_{bp}$ to POC conversion and the spatio-temporal variability in 10-d bins.

## Optical sediment trap methodology

The Optical Sediment Trap (OST) method uses the rate of change of particle attenuation (ATN = −ln(transmittance), in $m^2\,m^{-2}$) measured by the transmissometer during the float drifting period at the parking depth of 300 m[24,26,55]. The accumulation of small slow-sinking particles onto the upward-facing window of the transmissometer drives a smoothly increasing ATN while the accumulation of large fast-sinking particles produces discontinuities in transmissometer records. Both signals were converted into a continuous and a pulsed ATN flux (in units of $m^2\,m^{-2}\,d^{-1}$), respectively, following procedures described in Estapa et al.[26,55] with few modifications to take into account the use of a different platform with different parking behaviour and sampling frequency (every 1 h in this study). First, parking phases of duration <24 h were not considered to ensure enough data points. The first 3 data points (3 h) were removed from the analysis as the float takes time to stabilize at the target depth. Optical spikes, defined as an increase in ATN of 0.002 $m^2\,m^{-2}\,h^{-1}$ followed by a decrease within 3 h, were also removed and a 5-point median filter was applied. Then, the remaining data points were divided into linear segments interspersed with discontinuities, or 'jumps', following the procedure in Estapa et al.[55], except that discontinuities were not identified a priori with a threshold criteria but resulted from the subdivision of segments not meeting the fitting criteria. For each parking phase, the continuous flux of slow-sinking particles was computed as the mean slope of all linear segments weighted by their length. The pulsed flux was computed as the sum of all positive discontinuities normalized by the duration of the parking phase. Finally, both continuous and pulsed ATN fluxes were converted to POC flux (in mg C $m^{-2}\,d^{-1}$) following Estapa et al. (2019):[22]

$$F_{OST} = \langle F_{ATN} \rangle * 2133(\pm 173) + 1.4(\pm 5.6)$$

where $\langle F_{ATN} \rangle$ is the mean of either the continuous or the pulsed attenuance flux in 10-d bins. Values in parentheses are the standard deviations of the parameters of the linear fit. Note that this relationship is based on measurements made in the North Atlantic and may be inappropriate in other basins with very different particle assemblages. A SO-specific relationship does not currently exist. As for the $F_{spike}$, the probability distribution of $F_{OST}$ was computed using a Monte Carlo approach, with uncertainties introduced by the ATN to POC conversion and the spatio-temporal variability in 10-d bins.

Figure 3b, c allows us to compare the two flux estimates of the BGP, namely $F_{OST}$ and $F_{POC\ spike}$. For the three blooms, the pulsed $F_{OST}$ and $F_{POC\ spike}$ were of similar order of magnitude, but the temporal dynamics were different, especially during the 2016–17 bloom when $F_{POC\ spike}$ peaked in early December, ~30 days before the pulsed $F_{OST}$. This difference is likely due to the difference in the depth of the measured flux (100-m bin below the mixing layer versus the parking depth of 300 m). Indeed, the flux of fresh aggregates recorded in $F_{Chl\ spike}$ and $F_{POC\ spike}$ was quickly attenuated with depth (Fig. S6), and was therefore less intense at 300 m compared to the flux following the coccolithophore bloom in early January.

## Mixed layer pump

The methodology to estimate the magnitude of the mixed layer pump (MLP) was inspired by Dall'Olmo et al.[13] and Bol et al.[14]. The stock of entrained/detrained POC due to the change in the mixing layer depth

between two subsequent time steps was calculated as follows:

$$
\int POC(t) = \begin{cases} + \int_{\max(MLD_{bio}(t+1),100\,m)}^{MLD_{bio}(t)} POC(z,t)\,dz & \text{if } MLD_{bio}(t) > MLD_{bio}(t+1) \\ & \text{and } MLD_{bio}(t) > 100\,m \\ - \int_{\max(MLD_{bio}(t),100\,m)}^{MLD_{bio}(t+1)} POC(z,t)\,dz & \text{if } MLD_{bio}(t+1) > MLD_{bio}(t) \\ & \text{and } MLD_{bio}(t+1) > 100\,m \end{cases}
$$

In this calculation, $\int POC$ is positive when the mixing layer shoals ($MLD_{bio}(t) > MLD_{bio}(t+1)$) and POC stock is detrained from the mixing layer. The layer delimited by $MLD_{bio}(t)$ at the bottom and $MLD_{bio}(t+1)$ at the top is defined as the remnant layer (Fig. S11). Conversely, $\int POC$ is negative when the mixing layer deepens and POC stock is entrained from deeper waters. Following Dall'Olmo et al.[13], POC stocks from the surface to 100 m were not included in the calculation to avoid accounting for POC produced below the mixing layer, typically in summertime stratified waters. The nominal depth of 100 m corresponds approximately to the base of the euphotic layer in high latitude regions and is also consistent with other studies estimating carbon export[5]. The sum of $\int POC$ over a given period, divided by the duration of the period, provides an estimate of the net carbon export out of the productive layer $F_{MLP}$, expressed in mg C m$^{-2}$ d$^{-1}$. This approach assumes a spatial homogeneity in physical and biogeochemical properties over the area covered by the float displacement during the given period[56]. To smooth out any (sub)mesoscale spatial heterogeneities, $F_{MLP}$ was computed using a 20-d centred moving sum, and then averaged over 10-d bins to be consistent with flux estimates from the other pathways. Following a Monte Carlo approach, this process was repeated 5000 times, with uncertainties introduced by the $b_{bp}$ to POC conversion. Error bars in Fig. 3 highlight the uncertainties associated to both the conversion to POC and the spatio-temporal variability in 10-d bins.

## Eddy subduction pump

To detect subsurface features resulting from ESP events, we used an updated version of the method described in Llort et al.[9] and Omand et al.[8]. This method relies on the observation that the vertical extent of submesoscale features that drive ESP is of the order of ~10 m and can be smoothed out by averaging the float's vertical variability over larger (i.e., ~100 m) vertical scales. We applied a 20-bin centred running median over single vertical profiles interpolated at 5 m vertical resolution. As a result, we obtained submesoscale-free vertical variability that can be compared to the original profiles to identify anomalous features. We focused only on anomalies found between the bottom of the mixing layer depth and 500 m. As anomalies can be related to other mechanisms than submesoscale vertical circulation, individual profiles were classified as an ESP event if negative anomalies of Apparent Oxygen Utilisation (AOU') and spice (π') coincided in depth with a positive POC anomaly (POC'). If more than one anomaly was detected in the same profile, we only considered the shallowest one. The running median was computed with a centred filter, i.e., −/+10 bin median for each bin depth. To define the relevant anomalies, we applied detection thresholds to only consider anomalies with AOU' < −5 µmol kg$^{-1}$, POC' > 0 and π' < −0.05. Tests with positive spiciness anomalies (i.e., π' > 0.05) resulted in no new events detected. This result is likely because, in the Southern Ocean, spiciness is negative at the surface and decreases with depth[57], unlike other regions such the North Atlantic where Omand et al.[8] found positive π' anomalies associated with ESP events. We have also modified the method to better constrain the thickness of the anomalous features, an important metric to estimate the amount of POC exported. Here, we detected the top and bottom depths of both AOU' and π' as the first and last bin depths where the anomalies were still negative. We then compared the detected depths on AOU' and π' to define the deepest (shallowest) of the two as the anomaly bottom (top) depth.

For each detected feature, the associated vertical C export was computed as follows:

$$
F_{ESP} = POC_{surf} * \langle w_{sub} \rangle
$$

where $\langle w_{sub} \rangle$ is the mean vertical velocity of the injected water parcel, defined as:

$$
\langle w_{sub} \rangle = (MLD_{bio} - Z_{ev})/\Delta t
$$

where

$$
\Delta t = (POC_{surf} - POC_{ev})/R
$$

$R$ is the respiration rate of organic matter in the mesopelagic and is equal to $0.33(\pm 0.16)\,mg\,C\,m^{-2}\,d^{-1}$, the mean (and standard deviation) respiration rate between 100 and 400 m estimated from BGC-Argo floats in Southern Ocean open waters (Hennon et al.[58], their Figure 4). $POC_{surf}$ is the concentration of POC in the mixing layer averaged over all float profiles in the 10-d bin containing the detected subsurface feature. The standard deviation of POC concentration was also used to assess the spatio-temporal variability of $POC_{surf}$. Likewise, the mean and standard deviation of $MLD_{bio}$ were calculated in the same 10-d bin. $Z_{ev}$ and $POC_{ev}$ are the median depth and POC concentration of the subsurface feature, respectively. Following a similar approach as for the other export pathways, the probability distribution of $F_{ESP}$ was computed using a Monte Carlo simulation, with uncertainties introduced by the respiration rate R, the $b_{bp}$ to POC conversion and the spatio-temporal variability of $POC_{surf}$ and $MLD_{bio}$. This process was repeated for each subsurface feature detected in 10-d bins. Figure 3 shows the resulting median and interquartile range of $F_{ESP}$ for each 10-d bin. Note that $F_{ESP}$ is a gross export and does not include potential upward fluxes associated with ESP events. This approach relies on 4 assumptions:

1. The probability distribution of $POC_{surf}$ was representative of the $POC_{ev}$ concentration in the water parcel before being subducted.

2. The difference between $POC_{surf}$ and $POC_{ev}$ was only due to respiration of organic matter since the water parcel left the surface mixed layer.

3. The average respiration rate in the water parcel was constant in the water parcel.

4. The distance travelled by the water parcel since leaving the mixing layer until being detected was equivalent to the difference between the depth of detection $Z_{ev}$ and $MLD_{bio}$.

Note that all ESP features were detected much deeper than MLP remnant layers resulting in no double counting of physically driven POC export by the MLP and ESP (Fig. S14).

## Iron stress index

The phytoplankton iron stress index was computed following the method in Ryan-Keogh & Thomalla[36]. The concept underpinning this method is that NPQ variability is linked to both iron and light availability and has the potential to provide important diagnostic information on phytoplankton physiology[59]. To remove the effect of in situ light availability on NPQ variability, Ryan-Keogh & Thomalla[36] proposed to compute $\alpha_{NPQ}$ the initial slope of the NPQ-PAR curve. Thereby, $\alpha_{NPQ}$ could be used as a proxy for iron limitation, with higher values being associated with greater phytoplankton iron stress. In our study, NPQ as a function of depth was quantified as the difference between the quenching corrected fluorescence profile and the quenched one, normalized by the latter. For each profile, we plotted our iPAR estimates against NPQ values. We then applied a linear fit to the NPQ-iPAR curve in the region of low iPAR values (between 15 and 75 µmol photons m$^{-2}$ s$^{-1}$), where the slope of the linear fit gives $\alpha_{NPQ}$. Linear fits with $R^2 < 0.8$

were rejected (1% of the profiles). We did not fit a Platt-like model as in Ryan-Keogh & Thomalla[36] because our iPAR values were too low to induce a saturation plateau and because we were interested only in the initial slope of the NPQ-iPAR curve. Note that $\alpha_{NPQ}$, as a proxy for iron limitation, has to be interpreted with caution as shifts in phytoplankton community composition and changes in the light regime and thus phytoplankton photoacclimation status impact the variability of $\alpha_{NPQ}$[60,61]. For that reason, we also computed the median light level within the mixing layer ($I_{ML} = PAR_{SAT} e^{-0.5 K_{d(PAR)} MLD_{bio}}$) which is often used in photoacclimation models[62]. Here, $PAR_{SAT}$ is the daily mean PAR from MODIS Aqua (4 km) and $K_{d(PAR)}$ is the diffuse attenuation coefficient of PAR (in units of $m^{-1}$). $K_d$ was first computed at 490 nm by fitting a fourth-degree polynomial function to the logarithm of the downwelling irradiance $E_{d(490)}$ as a function of depth, measured by the float, and then calculating the mean slope over the first 50 m[63,64]. $K_{d(490)}$ was then converted to $K_{d(PAR)}$ following equation 9 in Morel et al. (2007)[65]. We did not find a clear relationship between $I_{ML}$ and $\alpha_{NPQ}$ (Fig. S4). The influence of shifts in phytoplankton community composition on $\alpha_{NPQ}$ variability cannot be ruled out, but phytoplankton community composition also shifts in response to iron availability[66].

### Phenology metrics

Phytoplankton phenology can be characterised by three metrics:[67] (i) the bloom onset when phytoplankton carbon stock starts accumulating (i.e. when the accumulation rate r changes from negative to positive), (ii) the bloom climax when r reaches its annual maximum value, (iii) the bloom apex when phytoplankton carbon stock P reaches its annual maximum value. Following Uchida et al.[68], P was estimated by vertically integrating $C_{phyto}$, derived from the backscattering at 470 nm, over the whole water column. To avoid including non-phytoplankton particulate matter (e.g. refractory material) in the calculation, $C_{phyto}$ was masked out at depths where $Chl_s = Chl - CHl_l - Chl_r \leq 0$, where $Chl_r = 0.3 \, mg \, m^{-3}$ (Fig. S9). The timeseries of $C_{phyto}$ was first linearly interpolated on equally spaced 5-day timeseries, and then smoothed with a 30-day (7 points) running average to filter out short-term fluctuations and focus on seasonality. The accumulation rate was then calculated as $= \frac{1}{P} \frac{\delta P}{\delta t}$. The bloom onset and apex were used as milestones to compare the timing of the 2016–17 and 2017–18 blooms with the seasonality of the carbon export pathways (Figs. S10 and S13). The time axis in Fig. S13 was rescaled by the onset and the apex of the bloom, so that 0 corresponds to the onset and 1 to the apex.

### Quasi-Lagrangian framework of the float trajectory

During its 36-month mission, the float visited different oceanic provinces and crossed strong water mass boundaries, such as in September 2016 when the float crossed the Polar Front. Thus, observed changes in biogeochemical properties were not solely due to temporal changes, confounding the study of the seasonality of these properties. We therefore divided the timeseries into three periods in which the contiguous nature of the water masses was verified based on temperature and salinity properties (24-Sep-2016 to 22-May-2017; 10-Oct-217 to 7-Jun-2018; 5-Nov-2018 to 1-May-2019; Fig. S2). The absence of strong water mass contrasts allows us to assume a quasi-Lagrangian framework, where changes in biogeochemical properties can be interpreted as temporal changes. This approach is commonly used in float studies[69–71]. For completeness, the Figures show the full float timeseries. However, we focus the Discussion on the three quasi-Lagrangian periods where we can confidently interpret the relationship between phytoplankton phenology in the surface, and the export and fate of POC in the underlying mesopelagic zone. All carbon flux estimates from the different export pathways remain robust for the whole timeseries.

## Data availability

BGC-Argo data used in this study can be downloaded from the Argo Global Data Assembly Center (ftp://ftp.ifremer.fr/ifremer/argo/). These data were collected and made freely available by the International Argo Program and the national programs that contribute to it: (http://www.argo.ucsd.edu, https://www.ocean-ops.org). The Argo Program is part of the Global Ocean Observing System. Ssalto/Duacs altimeter product was produced and distributed by the Copernicus Marine and Environment Monitoring Service (CMEMS, https://marine.copernicus.eu/). ERA5 wind speed and net heat flux reanalysis products were downloaded from https://cds.climate.copernicus.eu/cdsapp#!/home. Monthly GlobColour products were downloaded from CMEMS and MODIS products from the NASA Ocean Color website (https://oceancolor.gsfc.nasa.gov). The sea-ice product was downloaded from https://nsidc.org/data/nsidc-0081.

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

## Acknowledgements

L.L. was supported by a European Union's Horizon 2020 Marie Skłodowska-Curie grant (no. 892653). Visits by L.L. to IMAS/UTas were supported by the Australian Research Council Special Research Initiative for Antarctic Gateway Partnership (Project ID SR140300001), the Australian Research Council Centre of Excellence for Climate System Science (Project ID B0018492), and the UTas visiting scholar program. J.L. was supported by the European Union's Horizon 2020 research and innovation programme under the Marie Skłodowska-Curie grant agreement No. 754433. N.B. was supported by a European Research Council Consolidator Grant (GOCART, agreement number 724416). P.S. was supported by the Australian Research Council Centre of Excellence for Climate Extremes (Project ID CE170100023). P.W.B. was funded by the Australian Research Council by Laureate Fellowship FL160100131. We thank the Argo Data Management team (ADMT) and the BGC-Argo Data Management team (BGC ADMT). The specific float used in this study was funded by the ARC SRI for Antarctic Gateway Partnership (Project ID SR140300001). We thank the captain and crew of RV Investigator voyage IN2016_V03 and chief scientists Bernadette Sloyan and Susan Wijffels for deploying the float.

## Author contributions

L.L. and J.L. designed the study and conducted the data analysis. L.L. wrote the manuscript. J.L., P.S. and P.W.B. designed the float mission and contributed to manuscript review and editing. N.B. conducted the mean particle diameter analysis and contributed to manuscript review and editing. P.S. and P.W.B. helped to design the study.

## Competing interests

The authors declare no competing interests.
