## [Peer Review File · Nature Communications]

REVIEWER COMMENTS

Reviewer #1 (Remarks to the Author):

General assessment: The aim of this study is to characterize the seasonality and pathways of biogenic carbon export to the deep ocean in the Southern Ocean using observations from a single biogeochemical-Argo float. The authors compare the seasonality in the inferred passive gravitational settling of biogenic particles as well as particle “injection” from eddies and seasonal changes in mixed layer depth. The general scientific scope of this article is interesting and might be appealing to a wider public if some major modifications/improvements are included. While I was initially disappointed that this study is based on the results from only one single float, the selected float does have a wide suite of sensors that may permit a thorough assessment of the export of biogenic particles from the surface ocean. The float also sampled different environmental regimes with potentially different drivers of carbon export. However, I believe that the title of the manuscript should be modified to reflect the fact that the results only proceed from the Pacific basin and are not representative of entire Southern Ocean. A more important concern that I have with the current manuscript is that it does not provide a clear comparison between the magnitude (units of $\text{gr C/m}^2 / \text{time}$) of the different export mechanisms (pumps). As it currently stands, the “biological gravitational pump” is not properly estimated, and only a relative comparison between the timing of constant and pulse fluxes is provided (in units of day^{-1}). In line 164-166 the authors state that the measurements “are not calibrated in terms of carbon biomass”. Is there way to provide at least an approximate estimate with uncertainty bands? Similarly, figure 3d is missing temporal units. Is that correct or just a typo? Personally, I do not find very convincing the role of the mixed layer as a particle injection pump (PIP) since the dissolved inorganic carbon from respired particles detrained during summer can be re-entrained during winter mixing, resulting in no effect on air-sea CO_2 flux. In order for any ocean carbon pump to influence atmospheric CO_2 levels (as framed by reference 2 in the introduction (Volk and Hoffert, 1985)), carbon needs to be, at a minimum, exported below the maximum winter mixed layer depth. Hence, a direct way of proving the validity of the PIPs considered in this study would be to provide clear estimates of export fluxes from each of the pumps and integrating them through the annual growth cycle. Such an analysis would permit a real assessment of the role of each of the export pathway (BGP vs PIPs). While most of the manuscript is well written, I would recommend adding significantly more detail to the methods section, particularly as it refers to the computation of fluxes from the PIPs. Presently, it is difficult to understand all the calculations/assumptions involved in these estimates. I do not think that it is sufficient to just refers to previous works regarding the methods employed to estimate the eddy and mixed layer pumps, especially when the authors state that these methods have been refined (line 64–65). I suggest breaking the continuous narrative of the methods into steps with clear equations (and units) that ultimately lead to inferred estimates of downward carbon flux by each of the export pathways (gravitational settling, eddy, and mixed layer injection). Specific comments: Line 33: “It removes 3 PgC from the euphotic zone (Ez, upper ~ 100 m) annually, representing 33% of the global BCP1.” You might want to consider also citing similar but more recent estimates (Arteaga et al., 2018). Line 63: References 8 and 17 are the same. Line 66: “Concurrent measurements of chlorophyll a fluorescence and particulate backscattering (proxies of phytoplankton biomass and POC, respectively)”. Remove “respectively”. Phytoplankton biomass can also be estimated from particulate backscattering. Line 101: “and chlorophyll a fluorescence (F_s around 1 mg m^{-3} chlorophyll, Fig. S7)”. You say chlorophyll

fluorescence but provide concentrations units. This was confusing throughout the manuscript and in the figures. Please clarify if you are reporting fluorescence or concentration and use adequate units. Line 111: “potentially explaining the massive invasion of POC to 450 m following the coccolithophore bloom (Fig. 2a)”. This invasion is not clear. I would recommend plotting $\log(\text{POC})$ to better appreciate vertical and horizontal gradients, but use the actual value in the colorbar labels. Line 124: “The transition of the bloom to a subsurface chlorophyll maximum (SCM) feature was another indication of nutrient limitation. High F_s/bbs at the SCM (around ~ 125 m depth, Fig. 2b) was due to photoacclimation as light became limiting in the subsurface”. This is not clear. The SCM is observed in the fluorescence/ bbp ratio but not in the POC profiles. This would imply photoacclimation but not nutrient limitation. Line 164: “These measurements are however limited to a single depth, the float ‘parking’ depth, here ~ 300 m, and are not calibrated in terms of carbon biomass, so they are expressed in d^{-1} 19,34.”. Is there any way around this? Could you at least infer estimates of these carbon fluxes? Line 200: “delimited by MLD_{bio} at the top and MLD_{dens} at the bottom”. Shouldn’t it be MLD_{dens} at the top? This is what occurs during detrainment. Adding clear steps/equations to the methods can help avoid confusion. Line 223: “The contribution of all these processes over a complete annual cycle should definitely be accounted for when computing regional mesopelagic carbon budgets”. I agree! This is what is missing in this study. Line 292: Sometimes you used “ bbp ” and others “ bbs ”, what is the difference? I do not think C_p has been defined in the text before this point. Figure 1: MLD_{bio} and isopycnal lines are difficult to see. Figure 2c. It would be helpful to use Y-right axis to add the iron limitation diagnostic from figure S5. Figure 3. Please use different, contrasting colors (e.g., blue/red), for the bars in panels b) and c). Green bars on panel 3d are not visible (line 477). Figure S7. Consider plotting $\log(\text{Chl})$ and $\log(\text{C}_{phyto})$. The use of “F” to display chl concentration is confusing. References: Arteaga, L., Haeëntjens, N., Boss, E., Johnson, K. S., & Sarmiento, J. L. (2018). Assessment of export efficiency equations in the Southern Ocean applied to satellite-based net primary production. *Journal of Geophysical Research: Oceans*, 123, 2945–2964. <https://doi.org/10.1002/2018JC013787> Volk, T., & Hoffert, M. (1985). Ocean carbon pumps: Analysis of relative strengths and efficiencies in ocean-driven atmospheric CO_2 changes. In E. Sundquist and W. Broecker (Eds.), *The carbon cycle and atmospheric CO_2 : Natural variations Archean to present*. Chapman conference papers, 1984 (Geophysical Monograph 32, pp. 99–110). Washington, DC: American Geophysical Union.

Reviewer #2 (Remarks to the Author):

This article presents an 3 year long time-series of measurements from a profiling float in the Southern ocean that is used to assess different pathways for the export of particulate organic matter over the annual cycle. The suite of bio-optical measurements (and analyses that has been pioneered by the authors in previous work) reveal variability amongst the different particulate carbon pools that is interpreted as representative the phenology of export fluxes associated with phytoplankton blooms. These impressive findings are indeed worth publishing. But I have several criticisms of the article that are laid out below and hope that the authors will address these and improve the manuscript.

The article is descriptive without a clear message or conclusion. So the reader is left without a take-away. One reason is that the different mechanisms are evaluated using different metrics and the relative strengths of the different mechanisms cannot be compared (Fig. 3). Is it possible to normalize the measures in some way?

A second issue is that the bio-optical signals are interpreted without any in situ data, but based on relationships from the literature, so much of the interpretation is conjectured and sometimes tenuous. Also, there are uncertainties in the interpretation (for example, POC is based on bbp-700) and these need to be better characterized.

There is an impreciseness in many of the statements, which needs to be cleared up. For example, biomass is used interchangeably with POC, which is used interchangeably with bbp(700nm). Many of the details comments point this out.

Accessibility to a wider audience:

The article needs to do a better job of introducing the biological pumps that are being evaluated, in particular, the ML pump and the eddy-subduction pump, so that the article would be of interest to someone unfamiliar with the former literature about these mechanisms.

The excessive use of acronyms throughout the paper makes it difficult to read.

Terminology (or methods) are used without any introduction. For example, the spike index, POC, iron limitation index, F_I and bbp_I, as well as F_s, bbp_s are not attributed to any specific size or criteria and one has to look through the Methods.

The term “pump”, which is used throughout, is a misnomer. It suggests something quite different than how these mechanisms actually work. Even though it has been used in previous literature, I would suggest replacing it by “mechanism” or another more accurate word, to prevent the incorrect propagation of physical concepts in the biological literature (e.g. “eddy pumping”). Perhaps, terms that are well-established, like MLP, can be kept, but clarified, but calling all these mechanisms as “pumps” seems inappropriate.

Referencing of literature and scientific background.

The authors need to take more care in referencing the literature where appropriate, and with more accuracy. For example, there are several places in the Introduction where the text refers to previous work, without citing it.

Line 18: Multiple strands of evidence

Line 20: Recent model estimates ...

Line 24: recent advances in optical

Line 36: reference (3) is not appropriate here

Line 36: The text says “current estimates” ... but the reference is old

Some claims made in the introduction are not accurate. For example,

Line 23 (and repeated on Line 56): “little is known about the seasonality of their fluxes” Previous work on eddy-driven subduction (Omand et al) and Mixed layer pump has clearly pointed out the seasonal dependency of these mechanisms, even providing time-dependent flux estimates, or pointing out the importance of the seasonal transition. So this is not new understanding, but the value of the current manuscript is that it provides a year-round assessment of the mechanisms by combining several kinds of measurements.

Another example is on Line 53: “Boyd et al ... suggested...” is incorrect. The seasonality of these mechanisms was suggested in the papers where the mechanisms were proposed.

Background: Lines 40-52 : This paragraph needs to provide a better explanation of the ML and ES mechanisms for the unfamiliar reader, with specificity in citing of literature (rather than lumping together as in xxx)

Imprecise wording is chosen throughout. Some examples:

The term POC is used loosely throughout— without explaining that it is an estimate (based on bbp). What size range of particles does POC encompass?

Line 116 : this is confusing. Should it have said “when light is limiting”? It is when light is limiting that one can expect higher chl/C ratios. What is the C:Chl ratio here?

Line 118: Iron limitation index - this has a lot of caveats and needs better explanation before it is applied to the interpretation

Line 119-120: should the consumption of iron lead to higher or lower phytoplankton biomass?

Line 123: high mean particle size — explain how this is known

Line 124: the SCM may have been advected from another region

Line 125: Why is this an indication of nutrient limitation?

Line 131: high iron limitation index - explain how this is known

Line 133: biomass - could include much more than what is meant here

Line 135: Why “accordingly”?

Line 141: explain the spike index

Line 142: How does it quantify the abundance of large particles?

Line 144-5 F_l is not a subset of bbl

Line 146 which 2 components?

Line 148 precedes (instead of precede)

Line 149: where what the other study - and why should the same happen here?

Line 150: a similar phenology would explain ... (but this is an assumption and should be stated)

Line 153 - resulted is too strong - replace with: can be attributed to

Line 154 : explain that this interpretation assumes a similar behavior as was observed elsewhere

Line 169: delete well

Line 174: what do the two speeds correspond to in the coccolithophore blooms?

Line 176: flux of what?

Line 178-9: this conjecture (of disaggregation and fragmentation into smaller particles) is not backed up with any evidence

Lines 182-183: this is a lot to say based on the measurements (it seems like an over-interpretation of the data)

Line 209: Ez - euphotic zone. Wouldn't the large particles be exported by sinking?

Line 212: What do you mean by “could take over downward export”?

Lines 231-234: This was examined by Dever et al. GBC, 2020

Line 238: The floats are not process focussed

Line 242: delete “minimal”

Line 365: averaging the...

Lines 361 onward: the detection of subsurface features in this way predates Llort et al. It is not the “subduction events” that are detected, but it is the resulting subsurface features that are detected.

Line 375 version of what?

Calculation of spice (Fig. S2) — How are the 10 and 100 point filters applied? Does this refer to 10 or 100 profiles, or 10 or 100 points in a single profile? Spice should be in sigma (density) space, so how does

this relate? Shouldn't the spice be calculated for each water mass type separately? What is the resolution of the T,S data, and is it binned every meter?

The term "vertical section" (e.g. caption to fig S7) is conventionally used to denote a "slice", rather than a series of profiles along a drift. Replace the term with "vertical profiles".

Other points:

Paragraph starting Line 98: POC is introduced without explaining that is a derived quantity or saying how it is derived (which is in Methods)

Line 71: This novel combination - Avoid using such adjectives like "novel" and let the reader judge the work. Which novel combination are the authors referring to ?

Line 103: how small? (what size - is "large" and "small"?)

Line 102: here a chlorophyll-a value is given, but elsewhere the data is presented in terms of F_s and F_l ? Why? Similarly POC values are given, but bbp_s and bbp_l are also used. Explain why.

Overall, I think these are valuable observations that should be published after revision of the manuscript.

-Amala Mahadevan

Reviewer #3 (Remarks to the Author):

General comments:

The authors evaluate the magnitude of different carbon pumps (mixing, eddy and gravitational pumps) using recent advances in the field of optical biogeochemical measurements. The authors present a very nice dataset and do a great job at leveraging the wide array of measurements at hand (POC proxies, particle size proxy, subduction proxies, + satellite data on PIC, iron). Quantifying the contribution of the physical carbon pumps is a major question in the ocean community that has implications for our understanding of the global carbon cycle. This manuscript definitely brings novel and quantitative findings to the table on this topic and is relevant to the broad audience of Nature Communications. There are a few concerns that the authors should address to strengthen the manuscript before publication. These concerns fall into two main categories:

I) the paper crucially needs a more in-depth discussion of the key points that puts the results in perspective of prior work. In some cases, prior work aligns with the present results but in others they are or at least appear to be in contradiction. This includes the key point of the role of the eddy pump, only detected during one of the three blooms here, but hypothesized as a major pathway in prior work (e.g. Omand et al Science et al 2015; Boyd et al Nature 2019). See detailed comments #1,2,6,7,10,11.

II) the paper also needs more information on the physical pumps calculation (e.g. possible double counting, implications of having a subsurface chlorophyll maximum). This is required to convince the reader that the pumps are properly accounted for. See detailed comments #3-5,8,9,12,13.

Main Comments

1) Bloom description in main text

The first two blooms are relatively well described, i.e. discussion of F_s/B_b ratio, iron limitation and particle size in relationship to the bloom phenology (Cocco in 2016-17, diatmos in 2017-18).

a) The description of the third bloom however is purely descriptive and lack some interpretation for the reader. How do you interpret the differences in iron, particle size etc? What does this mean for the bloom itself. Please clarify?

b) Also in L 171-174: Can you explain what the pulsed and continuous flux are thought to represent for the coccolithophore bloom as you do with the diatom bloom?

2) Bias tied to the presence of coccolithophores.

The impact of biominerals in the first bloom is discussed in the main text as a potential factor that explains the increase in export (prevent remineralization, ballast). However, in the Method section points out that the POC calculation might actually be biased high because of the presence of biominerals see L 268 which reads:

“Note that the presence of coccolithophores during the 2016-17 bloom elevates bbp without necessarily elevating POC or C_{phyto} , potentially leading to overestimation of these carbon estimates.”

Please add discussion about the potential bias tied to your method to derive POC in the presence of Coccolithophores.

3) Crucial need to expand on the pump calculation: MLP and ESP double counting?

The method to compute the Mixed layer pump (MLP) assumes that any POC between the mixing layer (defined by max in Chlorophyll gradient MLD_bio) and the mixed layer (defined by the max in density gradient MLD_dens) can be attributed to MLP (once a mean mesopelagic POC is removed). The eddy subduction pump (ESP) is quantified by identifying anomalies in POC at vertical scales <100m (i.e. departures from the vertically averaged POC using s20 bins of 5m resolution). The ESP detection is done between the base of MLD_bio and 500m which must often overlap with the zone where the MLP is calculated. Aren't you double counting the POC exported by the ESP in the MLP by doing so? You have a criteria on spiciness and AOU to detect ESP events but do you actually remove these from the MLP then?

I might be missing something here. More information is needed to convince the reader (see some suggestion in comment #5 below).

4) Negative spiciness anomaly

In this manuscript, and in prior work by these authors, subducted layers are identified by negative AOU and spiciness anomalies, and positive POC anomalies. Yet Omand et al. (2015) identify a subducted layer with positive spiciness anomaly. Is there something specific about the dynamics of the region that constrain subducted layers with positive POC anomalies to have negative spiciness anomalies? Is this something that was tested? Could the fact that evidence of ESP were only found in one region/bloom be due to how subducted layers are identified? Please clarify in the text.

5) I strongly advise the authors to add some material that illustrates the calculation of these different physical pumps at the same location/profile.

Figure S9 shows AOU, spice and POC in Sept 2016 but the spice anomaly is not shown (using filtered spice). Please expand on one of these figures to add the various terms used to compute the pumps and the resulting pumps: both MLP and ESP (MLD_bio, MLD_dens, AOU, mesopelagic value of POC that you remove for MLP etc.). This figure should be used to demonstrate how the calculation of ESP and MLP works and that there is no double counting.

6) Discuss why the eddy pump is only detected in 2016/17? How is this compatible with prior work and your own intro putting forward the ESP as a major contributor of the global POC export.

Figure S2b shows the strong spice anomaly signal in June 2018 but there is actually no export by the ESP at this period on Fig 3 and as stated in main text (L187: "such features were detected only during 2016-17")! Is this because there is no POC to export at that time? Could DOC be exported in some of these spiciness events although you don't detect POC export?

Why then do you use this event to illustrate the ESP method in Figure S2?

Do you think it is realistic that there is no subduction event at other times in such a dynamical regime as the Southern Ocean? This deserves some discussion, especially in the context of prior work including Boyd et al 2019, Omand et al 2015 and your own introduction that hypothesizes this pathway strongly contributes to carbon export. How can it be the case if for two years of sampling it is not detected? Could DOC export by these features bridge the gap of carbon export? Some discussion about this (apparent?) discrepancy is needed.

7) Discuss implications for PIPs of deep chlorophyll max in the case $MLD_{bio} > MLD_{dens}$

Please expand and discuss on the implications of the case where $MLD_{bio} > MLD_{dens}$ for your estimate of MLP and ESP.

L 358 in methods states: “..in the presence of SCM, typically in summer, MLD_{bio} can be deeper than the mixing layer depth and corresponds more likely to the base of the euphotic zone.”

a) Does that mean for MLP? IS $MLP = 0$ in this case?

b) What does this mean for the ESP? Can the eddy pump which is triggered by instabilities of MLD_{dens} or ageostrophic circulation reach down that deep if MLD_{dens} is shallower than MLD_{bio} ? Or do you actually find that ESP is close to zero in this case? If you don't find it to be zero, how do you physically explain the fact that the eddy pump would reach that deep?

Please add some material on the values of these pumps in this particular case. These are key questions for the community; your nice dataset might bring some answers even if they are partial answers. This discussion should probably be included in the main text.

8) Definition of the pseudo-Lagrangian framework

When the authors say that the float did not cross strong water mass boundaries during the period of interest, do they mean to say that each profile falls more or less along the same line in the T/S diagram? How was this determined? Can you show an example of when a float crossed a boundary? Maybe it would be easier to understand if the boundaries for the Pseudo-Lagrangian framework were added to the temperature and salinity time series (Fig. 1). Also, what does 'station number' (Fig. S3) mean? Is it the profile number?

9) Implications for pumps calculation during “non-Lagrangian” periods

Please clarify the implications tied to the definition of the three quasi Lagrangian periods in the method section L280. The method sections states: “For completeness, the figures show the full float timeseries, but only the three quasi-Lagrangian periods are discussed in the text.”

Does this mean that we cannot trust the pumps shown in figure 3 if they are computed between these periods? If this is the case, the times outside of these three periods should be shaded or hatched or similar in all figures so the reader can identify the periods that can be interpreted robustly. Implications for these “non lagrangian periods should be discussed”

10) How can the reader get a full picture of the three pumps?

The gravitational pump is in d^{-1} while other pumps are in gC/m^2 . How can the reader compare and interpret these fluxes (Figure 3)? Some sentences that discuss this point when the authors synthesize their findings are needed. For instance, L. 223-225: ‘The contribution of all these processes over a complete annual cycle should definitely be accounted for when computing regional mesopelagic carbon budgets’. This statement is vague, and considering the detailed analysis of multiple carbon pumps in this manuscript, the authors should discuss the relative contributions of the different pump in their region, at least the two physical pumps which are given in the same unit. I realize that the gravitational pump is not provided in terms of carbon flux, but perhaps the authors can discuss rough conversion estimates or what measurements would be necessary to constrain their findings.

11) Discussion, perspective and link to prior work

Some of your results lack a discussion of prior work to put them in perspective. Here are some important points to discuss:

- the timing of the MLP confined to spring restratification when biology is active and mixed layer is variable is consistent with model results from Resplandy et al 2019.

- possible interplay between PIP and BGP: you highlight that PIP might be more effective at injecting particles than the gravitational pump by passing remineralization (L210, 230). This is in apparent contradiction with Stukel et al 2017 who showed that sinking particles were more efficiently exported than smaller subducted particles that remained trapped in the upper water column. This difference in behavior might be related to the region (SO vs California Current). Note that the Dever paper you mention in your conclusion examine particle sizes that might not match the type of particles you are considering.

- the overall lack of ESP in your data after 2017 which seems in contradiction with a lot of recent works including Boyd et al Omand et al etc... (see details in comment #4).

12) Running average filter for eddy subduction pump

Fig S13 shows the depth of the eddy subduction pump. This is obtained from the filtered vertical profiles. Did the author apply a forward, backward, or forward-backward filter? We would expect a forward or a backward filter to shift the depth of the layer. How does the phase shift affect the results shown in Fig. S13?

13) 159-160 'In all three blooms, FI was rapidly attenuated with depth (...)' I agree with this statement for the blooms in 16/17 and 17/18, but find it debatable for 18/19. Can you speculate as to why there is a difference? The statement should be adjusted to refer to the first two blooms only.

Minor comments

L 130: "from December to February 2018". Do you mean December 2017 or December 2018 to Feb 2019?

Some sentences are formulated as statements although they are probably more hypothesis. For instance L153: "Low FI and maximum bbl spike indices in mid-January 2017 following the coccolithophore bloom resulted from sinking aggregates partly made of detached coccoliths³²."

Shouldn't this be rephrased with the idea that it is compatible with the expectation that the bloom is followed by an export of detached coccoliths? You don't actually have in-situ measurements that show these are coccoliths, do you?

L 202: 'All MLP events occurred prior to the bloom apex, each year (Fig. 3d ...): please label bloom stages on Fig 3 to support this statement (will also help support statement on L 218-219).

L 218-219: Remind readers that ESP were only observed in a highly energetic region (but see concerns regarding negative spiciness anomaly, comments # 4)

L 288: 'Smoothened' should be 'smoothed' for signal processing

L292: I don't think you define C_p in the text before using it in Equation 1 and where it comes from. You mention the beam attenuation later in the text (L333) but do not make a clear connection with the C_p

you use for the mean particle size computation. In the float description you mention b_{bp} but not C_p . Please clarify.

Fig. 3d: The empty bars are not green on my version; please adjust the figure caption

Mixed/mixing/remnant layers: I appreciate that the authors distinguish between the mixing and mixed layer. Currently, these concepts are only defined in the methods; please move the definitions earlier in the manuscript.

Diverging colormaps: The authors frequently make use of a diverging red/blue colormap for continuous variables. Unless the divergence is aligned with a meaningful threshold, this choice makes figures more difficult to interpret. Temperature, salinity, and month (Fig. 1); POC and F/b (Figs. 2, S3 & S11); and buoyancy frequency (Fig. S1) should all have continuous colormaps.

Resplandy, L., Lévy, M., McGillicuddy, D.J., 2019. Effects of Eddy-Driven Subduction on Ocean Biological Carbon Pump. *Global Biogeochem. Cycles* 33, 1071–1084. <https://doi.org/10.1029/2018GB006125>

Stukel, M.R., Song, H., Goericke, R., Miller, A.J., 2017. The role of subduction and gravitational sinking in particle export, carbon sequestration, and the remineralization length scale in the California Current Ecosystem: Subduction and sinking particle export in the CCE. *Limnology and Oceanography* 63, 363–383. <https://doi.org/10.1002/lno.10636>

Responses to Reviewers

We thank the three Reviewers for their constructive comments and suggestions which, we believe have considerably improved our manuscript. Here we present our detailed point-by-point responses and description of action taken in regards to the comments by each Reviewer. Reviewer comments are in regular text, responses are in blue and new text from the manuscript is in *blue italics*.

Reviewer #1:

Review of “Multi-year robotic observations reveal the seasonality of downward carbon export pathways in the Southern Ocean” by Lacour et al.

General assessment: The aim of this study is to characterize the seasonality and pathways of biogenic carbon export to the deep ocean in the Southern Ocean using observations from a single biogeochemical-Argo float. The authors compare the seasonality in the inferred passive gravitational settling of biogenic particles as well as particle “injection” from eddies and seasonal changes in mixed layer depth. The general scientific scope of this article is interesting and might be appealing to a wider public if some major modifications/improvements are included. While I was initially disappointed that this study is based on the results from only one single float, the selected float does have a wide suite of sensors that may permit a thorough assessment of the export of biogenic particles from the surface ocean. The float also sampled different environmental regimes with potentially different drivers of carbon export. However, I believe that the title of the manuscript should be modified to reflect the fact that the results only proceed from the Pacific basin and are not representative of entire Southern Ocean.

The new title is “Multi-year robotic observations reveal the seasonality of downward carbon export pathways in the Southern Ocean Pacific sector”

A more important concern that I have with the current manuscript is that it does not provide a clear comparison between the magnitude (units of $\text{gr C}/\text{m}^2/\text{time}$) of the different export mechanisms (pumps). As it currently stands, the “biological gravitational pump” is not properly estimated, and only a relative comparison between the timing of constant and pulse fluxes is provided (in units of day^{-1}). In line 164-166 the authors state that the measurements “are not calibrated in terms of carbon biomass”. Is there way to provide at least an approximate estimate with uncertainty bands? Similarly, figure 3d is missing temporal units. Is that correct or just a typo?

Those calculations and Figure 3 have been substantially revised. The different export mechanisms are now in the same carbon-based units of $\text{mg C m}^{-2} \text{day}^{-1}$ with uncertainties.

Personally, I do not find very convincing the role of the mixed layer as a particle injection pump (PIP) since the dissolved inorganic carbon from respired particles detrained during summer can be re-entrained during winter mixing, resulting in no effect on air-sea CO₂ flux. In order for any ocean carbon pump to influence atmospheric CO₂ levels (as framed by reference 2 in the introduction (Volk and Hoffert, 1985)), carbon needs to be, at a minimum, exported below the maximum winter mixed layer depth. Hence, a direct way of proving the validity of the PIPs considered in this study would be to provide clear estimates of export fluxes from each of the pumps and integrating them through the annual growth cycle. Such an analysis would permit a real assessment of the role of each of the export pathway (BGP vs PIPs).

We thank the reviewer for this suggestion. We now integrate the contribution of the pumps over a complete annual cycle. The net annual export by the MLP was 5.7 g C m⁻² in 2016-17 and 2.3 g C m⁻² in 2017-18 (not calculated in 2018-19 due to missing data for nearly two months), demonstrating that the MLP is an efficient pathway to transfer POC below the maximum winter mixing layer depth. However, a net POC export does not necessarily lead to CO₂ sequestration. The same calculation should be done with DIC and not POC to really estimate the role of the MLP on CO₂ sequestration. Unfortunately, we do not have DIC data. However, we want to stress here that remineralisation is not the sole fate for particles transported by PIPs. As discussed in the main text (section *Implications for oceanic carbon storage*), these particles may interact with other particles, such as the sinking assemblage associated with the Biological Gravitational Pump, and ultimately sink deeper, where sequestration time scales will be longer.

While most of the manuscript is well written, I would recommend adding significantly more detail to the methods section, particularly as it refers to the computation of fluxes from the PIPs. Presently, it is difficult to understand all the calculations/assumptions involved in these estimates. I do not think that it is sufficient to just refers to previous works regarding the methods employed to estimate the eddy and mixed layer pumps, especially when the authors state that these methods have been refined (line 64—65). I suggest breaking the continuous narrative of the methods into steps with clear equations (and units) that ultimately lead to inferred estimates of downward carbon flux by each of the export pathways (gravitational settling, eddy, and mixed layer injection).

The Methods section has been completely revamped to take into account the new methods used to calculate all fluxes in carbon-based units. The narrative of the Methods section has been broken down into clear steps as requested.

Specific comments:

Line 33: “It removes 3 PgC from the euphotic zone (Ez, upper ~100 m) annually, representing 33% of the global BCP1.” You might want to consider also citing similar but more recent estimates (Arteaga et al., 2018).

Done

Line 63: References 8 and 17 are the same.

Corrected

Line 66: “Concurrent measurements of chlorophyll a fluorescence and particulate backscattering (proxies of phytoplankton biomass and POC, respectively)”. Remove “respectively”. Phytoplankton biomass can also be estimated from particulate backscattering.

Done

Line 101: “and chlorophyll a fluorescence (Fs around 1 mg m⁻³ chlorophyll, Fig. S7)”. You say chlorophyll fluorescence but provide concentrations units. This was confusing throughout the manuscript and in the figures. Please clarify if you are reporting fluorescence or concentration and use adequate units.

We do not refer to chlorophyll fluorescence anymore. Now, all chlorophyll data are expressed in concentration units. Chl refers to total chlorophyll concentration while Chl_s and Chl_l refer to small-particle and large-particle chlorophyll concentration, respectively.

Line 111: “potentially explaining the massive invasion of POC to 450 m following the coccolithophore bloom (Fig. 2a)”. This invasion is not clear. I would recommend plotting log(POC) to better appreciate vertical and horizontal gradients, but use the actual value in the colorbar labels.

Done

Line 124: “The transition of the bloom to a subsurface chlorophyll maximum (SCM) feature was another indication of nutrient limitation. High Fs/ bbs at the SCM (around ~125 m depth, Fig. 2b) was due to photoacclimation as light became limiting in the subsurface”. This is not clear. The SCM is observed in the fluorescence/bbp ratio but not in the POC profiles. This would imply photoacclimation but not nutrient limitation.

This sentence has been removed.

Line 164: “These measurements are however limited to a single depth, the float ‘parking’ depth, here ~ 300 m, and are not calibrated in terms of carbon biomass, so they are expressed in d-1 19,34.“. Is there any way around this? Could you at least infer estimates of these carbon fluxes?

All fluxes are now expressed in mg C or g C m⁻² day⁻¹. See the new methods.

Line 200: “delimited by MLD_{bio} at the top and MLD_{dens} at the bottom”. Shouldn’t it be MLD_{dens} at the top? This is what occurs during detrainment. Adding clear steps/equations to the methods can help avoid confusion.

MLD_{dens} is not used anymore in the new calculation. Equations have been added in the methods section.

Line 223: “The contribution of all these processes over a complete annual cycle should definitely be accounted for when computing regional mesopelagic carbon budgets”. I agree! This is what is missing in this study.

We now integrate all export pathways over complete annual cycles.

Lune 292: Sometimes you used “bbp” and others “bbs”, what is the difference? I do not think Cp has been defined in the text before this point.

This has been clarified in the main text and in the Methods section.

Figure 1: MLDbio and isopycnal lines are difficult to see.

Corrected

Figure 2c. It would be helpful to use Y-right axis to add the iron limitation diagnostic from figure S5.

Done

Figure 3. Please use different, contrasting colors (e.g., blue/red), for the bars in panels b) and c). Green bars on panel 3d are not visible (line 477).

Done

Figure S7. Consider plotting Log(Chl) and Log(Cphyto). The use of “F” to display chl concentration is confusing.

Done

References:

Arteaga, L., Haeëntjens, N., Boss, E., Johnson, K. S., & Sarmiento, J. L. (2018). Assessment of export efficiency equations in the Southern Ocean applied to satellite-based net primary production. *Journal of Geophysical Research: Oceans*, 123, 2945–2964. <https://doi.org/10.1002/2018JC013787>

Volk, T., & Hoffert, M. (1985). Ocean carbon pumps: Analysis of relative strengths and efficiencies in ocean-driven atmospheric CO₂ changes. In E. Sundquist and W. Broecker (Eds.), *The carbon cycle and atmospheric CO₂: Natural variations Archean to present*. Chapman conference papers, 1984 (*Geophysical Monograph* 32, pp. 99–110). Washington, DC: American Geophysical Union.

Reviewer #2 (Remarks to the Author):

This article presents an 3 year long time-series of measurements from a profiling float in the Southern ocean that is used to assess different pathways for the export of particulate organic matter over the annual cycle. The suite of bio-optical measurements (and analyses that has been pioneered by the authors in previous work) reveal variability amongst the different particulate carbon pools that is interpreted as representative the phenology of export fluxes associated with

phytoplankton blooms. These impressive findings are indeed worth publishing. But I have several criticisms of the article that are laid out below and hope that the authors will address these and improve the manuscript.

The article is descriptive without a clear message or conclusion. So the reader is left without a take-away. One reason is that the different mechanisms are evaluated using different metrics and the relative strengths of the different mechanisms cannot be compared (Fig. 3). Is it possible to normalize the measures in some way?

All export pathways are now expressed in mg C or g C m⁻² day⁻¹ (see new Methods and Figure 3). We have added a paragraph in the discussion to compare the magnitude and seasonality of the fluxes over complete annual cycles.

A second issue is that the bio-optical signals are interpreted without any in situ data, but based on relationships from the literature, so much of the interpretation is conjectured and sometimes tenuous. Also, there are uncertainties in the interpretation (for example, POC is based on bbp-700) and these need to be better characterized.

While we agree that the field of BGC-Argo analysis and interpretation would be best advanced by presenting comparisons with co-located in situ measurements where possible, in virtually all cases this is not possible due to the geographical remoteness of where floats are deployed (such as the polar Southern Ocean where match-ups are rare). Here, we must rely upon robust, previously published relationships. Uncertainties associated with these relationships have been taken into account in the new calculation of carbon fluxes.

There is an impreciseness in many of the statements, which needs to be cleared up. For example, biomass is used interchangeably with POC, which is used interchangeably with bbp(700nm). Many of the details comments point this out.

We have clarified this in the text.

Accessibility to a wider audience:

The article needs to do a better job of introducing the biological pumps that are being evaluated, in particular, the ML pump and the eddy-subduction pump, so that the article would be of interest to someone unfamiliar with the former literature about these mechanisms.

We have added details in the introduction to better define the mechanisms, especially the MLP.

The excessive use of acronyms throughout the paper makes it difficult to read.

We have tried to reduce the use of acronyms as much as possible. Most of acronyms are redefined in figure captions to help the reader.

Terminology (or methods) are used without any introduction. For example, the spike index, POC, iron limitation index, F_l and bbp_l, as well as F_s, bbp_s are not attributed to any specific size or criteria and one has to look through the Methods.

We have tried to better introduce these terms in the main text. For example, specifying the limits for large and small particles. But we don't want to add too much detail in the main text to avoid breaking the flow. We have significantly expanded the Methods section with more detail on all of the parameters mentioned by the reviewer.

The term “pump”, which is used throughout, is a misnomer. It suggests something quite different than how these mechanisms actually work. Even though it has been used in previous literature, I would suggest replacing it by “mechanism” or another more accurate word, to prevent the incorrect propagation of physical concepts in the biological literature (e.g. “eddy pumping”). Perhaps, terms that are well-established, like MLP, can be kept, but clarified, but calling all these mechanisms as “pumps” seems inappropriate.

We discussed among the author group, the merits of pump vs pathway vs mechanism. We decided that historical use of the word pump – going back to the seminal paper in AGU Monographs by Volk and Hoffert in the 1980's – and subsequently by many groups such as Marina Levys makes it a readily understood and legitimate term. As a compromise, in places within the text we do interchangeably use the term ‘mechanisms’ or ‘pathways’ with ‘pump’.

Referencing of literature and scientific background.

The authors need to take more care in referencing the literature where appropriate, and with more accuracy. For example, there are several places in the Introduction where the text refers to previous work, without citing it.

Line 18: Multiple strands of evidence

Line 20: Recent model estimates ...

Line 24: recent advances in optical

All these sentences are in the abstract, which cannot contain references according to Nature Communication rules.

Line 36: reference (3) is not appropriate here

Reference 3 (now reference 4) is a review of present knowledge on the different export pathways, and also the first time where the term Biological Gravitational Pump (BGP) was introduced, as opposed to the generic term Biological Carbon Pump. We think this reference is thus appropriate here. We also added a reference to a more recent review of the different export pathways.

Line 36: The text says “current estimates” ... but the reference is old

We added a citation to back up this statement (Emerson 2014).

Some claims made in the introduction are not accurate. For example, Line 23 (and repeated on Line 56): “little is known about the seasonality of their fluxes” Previous work on eddy-driven subduction (Omand et al) and Mixed layer pump has clearly pointed out the seasonal dependency of these mechanisms, even providing time-dependent flux estimates, or pointing out the importance of the seasonal transition. So this is not new understanding, but the

value of the current manuscript is that it provides a year-round assessment of the mechanisms by combining several kinds of measurements.

We acknowledge that the seasonality of export mechanisms is not a new understanding. We added some references to previous studies that have alluded to the importance of this seasonality. However, these studies lacked the high resolution, multi-year and multi-sensor observations of the seasonal cycle that we present here.

Another example is on Line 53: “Boyd et al ... suggested...” is incorrect. The seasonality of these mechanisms was suggested in the papers where the mechanisms were proposed.

Agreed, we have amended the text to make this clear.

Background: Lines 40-52 : This paragraph needs to provide a better explanation of the ML and ES mechanisms for the unfamiliar reader, with specificity in citing of literature (rather than lumping together as in xxx)

We have modified this paragraph as suggested.

Imprecise wording is chosen throughout. Some examples:

The term POC is used loosely throughout— without explaining that it is an estimate (based on bbp). What size range of particles does POC encompass?

The term POC is now defined in the introduction. Information on particle size is given later in the text when we introduce the decomposition of the bbp signal into large and small components.

Line 116 : this is confusing. Should it have said “when light is limiting”? It is when light is limiting that one can expect higher chl/C ratios. What is the C:Chl ratio here?

These high Chl/ b_{bp} values were observed during spring diatom blooms in the North Atlantic (Cetinić et al., 2015) and in the Southern Ocean (Rembauville et al., 2017) when light was not limiting. Changes in the phytoplankton community composition also affect the Chl to b_{bp} ratio (not only photoacclimation). But we agree that the sentence was confusing, we removed ‘*when light is not limiting*’.

Line 118: Iron limitation index - this has a lot of caveats and needs better explanation before it is applied to the interpretation

This explanation has been improved in the rewriting of the manuscript.

Line 119-120: should the consumption of iron lead to higher or lower phytoplankton biomass?

It should lead to higher biomass. We agree that this sentence was confusing. It has been removed.

Line 123: high mean particle size — explain how this is known

A quick explanation of the method has been added in the figure caption. Details can be found in the Methods section.

Line 124: the SCM may have been advected from another region

We did not see any evidence of advection in temperature, salinity or spice profiles.

Line 125: Why is this an indication of nutrient limitation?

This sentence has been removed.

Line 131: high iron limitation index - explain how this is known

We added a quick explanation of the method in the main text and in the figure caption. Details can be found in the Methods section.

Line 133: biomass - could include much more than what is meant here

'*Biomass*' has been replaced with '*POC*'

Line 135: Why "accordingly"?

Now that the iron limitation index has been explained in the text, the reader should better understand this sentence.

Line 141: explain the spike index

We added a quick explanation of the method in the main text. Details can be found in the Methods section.

Line 142: How does it quantify the abundance of large particles?

We no longer use this spike index.

Line 144-5 F_1 is not a subset of b_{bl}

By '*subset*', we mean to say that particles that contribute to the F_1 signal also contribute to the b_{bl} signal. The reverse is not always true however. This sentence has been removed in the new version of the manuscript.

Line 146 which 2 components?

The methods have changed. We now calculate the gravitational fluxes driven by the large fluorescing ($F_{chl\ spike}$) and backscattering ($F_{POC\ spike}$) particles (i.e. the two components).

Line 148 precedes (instead of precede)

Line 149: where what the other study - and why should the same happen here?

Line 150: a similar phenology would explain ... (but this is an assumption and should be stated)
Line 153 - resulted is too strong - replace with: can be attributed to
Line 154 : explain that this interpretation assumes a similar behavior as was observed elsewhere

This paragraph has been totally rearranged. Most of the above-mentioned sentences have been removed.

Line 169: delete well

Done

Line 174: what do the two speeds correspond to in the coccolithophore blooms?

We no longer estimate sinking speeds.

Line 176: flux of what?

This sentence has been removed.

Line 178-9: this conjecture (of disaggregation and fragmentation into smaller particles) is not backed up with any evidence

Indeed, it is an assumption based on the delay in time between the peaks of small and large particle fluxes. This sentence has been reformulated.

Lines 182-183: this is a lot to say based on the measurements (it seems like an over-interpretation of the data)

This sentence has been reformulated.

Line 209: Ez - euphotic zone. Wouldn't the large particles be exported by sinking?

We think that a combination of both mechanisms is possible.

Line 212: What do you mean by "could take over downward export"?

This sentence has been reformulated

Lines 231-234: This was examined by Dever et al. GBC, 2020

Indeed, we have already cited this work in the text. Here, we show with in situ observations that the timing of ESP events with respect to phytoplankton phenology (i.e. particle characteristics) is favorable for the interplay with the BGP.

Line 238: The floats are not process focussed

This sentence has been removed. However, we do think that floats can be process focused with an appropriate sampling resolution.

Line 242: delete “minimal”

This paragraph has been removed.

Line 365: averaging the...

Corrected

Lines 361 onward: the detection of subsurface features in this way predates Llort et al. It is not the “subduction events” that are detected, but it is the resulting subsurface features that are detected.

Agreed, we have added the reference of Omand et al. (2015) and amended the text to make this clear.

Line 375 version of what?

We have reformulated this sentence.

Calculation of spice (Fig. S2) — How are the 10 and 100 point filters applied? Does this refer to 10 or 100 profiles, or 10 or 100 points in a single profile? Spice should be in sigma (density) space, so how does this relate? Shouldn't the spice be calculated for each water mass type separately? What is the resolution of the T,S data, and is it binned every meter?

This indeed refers to data points in a single profile, interpolated at 5-m resolution. We have clarified this in the text. Figure S2 has been removed. We don't think there is anything wrong in plotting spice profiles in pressure space. This has been done in many other studies, including Omand et al. (2015) and Llort et al. (2017).

The term “vertical section” (e.g. caption to fig S7) is conventionally used to denote a “slice”, rather than a series of profiles along a drift. Replace the term with “vertical profiles”.

Done

Other points:

Paragraph starting Line 98: POC is introduced without explaining that is a derived quantity or saying how it is derived (which is in Methods)

We clarified this in the introduction.

Line 71: This novel combination - Avoid using such adjectives like “novel” and let the reader judge the work. Which novel combination are the authors referring to ?

Agreed, we removed the word '*novel*'. The combination refers to the conjoint use of the methods described earlier in the paragraph.

Line 103: how small? (what size - is "large" and "small"?)

The theoretical size limit between small and large is $\sim 100 \mu\text{m}$. We have added this information in the text.

Line 102: here a chlorophyll-a value is given, but elsewhere the data is presented in terms of F_s and F_l ? Why? Similarly POC values are given, but bbp_s and bbp_l are also used. Explain why.

We initially used the terms F_s and F_l to be consistent with the study by Briggs et al. (2020) who first introduced this segregation between large and small particles, but we agree that it was confusing. We now refer to Chl_s and Chl_l in concentration units. The difference between POC, b_{bs} and b_{bl} is now better explained in the text.

Overall, I think these are valuable observations that should be published after revision of the manuscript.

Thank you

-Amala Mahadevan

Reviewer #3 (Remarks to the Author):

General comments:

The authors evaluate the magnitude of different carbon pumps (mixing, eddy and gravitational pumps) using recent advances in the field of optical biogeochemical measurements. The authors present a very nice dataset and do a great job at leveraging the wide array of measurements at hand (POC proxies, particle size proxy, subduction proxies, + satellite data on PIC, iron). Quantifying the contribution of the physical carbon pumps is a major question in the ocean community that has implications for our understanding of the global carbon cycle. This manuscript definitely brings novel and quantitative findings to the table on this topic and is relevant to the broad audience of Nature Communications. There are a few concerns that the authors should address to strengthen the manuscript before publication. These concerns fall into two main categories:

I) the paper crucially needs a more in-depth discussion of the key points that puts the results in perspective of prior work. In some cases, prior work aligns with the present results but in others they are or at least appear to be in contradiction. This includes the key point of the role of the eddy pump, only detected during one of the three blooms here, but hypothesized as a major pathway in prior work (e.g. Omand et al Science et al 2015; Boyd et al Nature 2019). See detailed comments #1,2,6,7,10,11.

We agree with the reviewer that the importance of ESP concerning the other mechanisms has not been well conveyed in the text. We have addressed this general comment in the specific questions below. We also added few sentences in the main text to discuss the relative importance of the ESP in light of previous work.

II) the paper also needs more information on the physical pumps calculation (e.g. possible double counting, implications of having a subsurface chlorophyll maximum). This is required to convince the reader that the pumps are properly accounted for. See detailed comments #3-5,8,9,12,13.

The methodology to estimate the magnitude of the export pathways has been significantly improved. All fluxes are now expressed in carbon-based units. The calculation of the physical pumps is now detailed with clear steps/equations. We also added 2 figures (Fig. S11 and S14) in the supplementary to tackle the issue of double counting.

Main Comments

1) Bloom description in main text

The first two blooms are relatively well described, i.e. discussion of Fs/Bbs ratio, iron limitation and particle size in relationship to the bloom phenology (Cocco in 2016-17, diatoms in 2017-18).

a) The description of the third bloom however is purely descriptive and lack some interpretation for the reader. How do you interpret the differences in iron, particle size etc? What does this mean for the bloom itself. Please clarify?

We voluntarily limited the discussion of the third bloom because data were missing for nearly two months early in the productive season and because the two peaks in phytoplankton biomass occurred in two different regions. Even if temperature and salinity data do not show a clear contrast between these two regions, the interpretation of the observations is more complex.

b) Also in L 171-174: Can you explain what the pulsed and continuous flux are thought to represent for the coccolithophore bloom as you do with the diatom bloom?

We did not find enough information on the phenology of coccolithophore blooms and characteristics of the associated export flux in the literature to fully interpret our OST results, as we have done for the diatom bloom, which is a more common and studied feature.

2) Bias tied to the presence of coccolithophores.

The impact of biominerals in the first bloom is discussed in the main text as a potential factor that explains the increase in export (prevent remineralization, ballast). However, in the Method section points out that the POC calculation might actually be biased high because of the presence of biominerals see L 268 which reads:

“Note that the presence of coccolithophores during the 2016-17 bloom elevates bbp without necessarily elevating POC or C_{phyto}, potentially leading to overestimation of these carbon estimates.”

Please add discussion about the potential bias tied to your method to derive POC in the presence of Coccolithophores.

We added a sentence in the main text to acknowledge this limitation.

3) Crucial need to expand on the pump calculation: MLP and ESP double counting?

The method to compute the Mixed layer pump (MLP) assumes that any POC between the mixing layer (defined by max in Chlorophyll gradient MLD_{bio}) and the mixed layer (defined by the max in density gradient MLD_{dens}) can be attributed to MLP (once a mean mesopelagic POC is removed). The eddy subduction pump (ESP) is quantified by identifying anomalies in POC at vertical scales $<100m$ (i.e. departures from the vertically averaged POC using s20 bins of 5m resolution). The ESP detection is done between the base of MLD_{bio} and 500m which must often overlap with the zone where the MLP is calculated. Aren't you double counting the POC exported by the ESP in the MLP by doing so? You have a criteria on spiciness and AOU to detect ESP events but do you actually remove these from the MLP then?

I might be missing something here. More information is needed to convince the reader (see some suggestion in comment #5 below).

The method to estimate the magnitude of the MLP totally changed. We no longer use the density gradient MLD_{dens} . The new figure S14 demonstrates that the penetration depth of particles is different between the two physically-mediated mechanisms. The detected subsurface features associated to ESP events were much deeper than the layer considered for the calculation of the MLP (see also Fig. S11). This could be different in other regions, but here, there is no double counting. This has been clarified in the Methods and in the main text.

4) Negative spiciness anomaly

In this manuscript, and in prior work by these authors, subducted layers are identified by negative AOU and spiciness anomalies, and positive POC anomalies. Yet Omand et al. (2015) identify a subducted layer with positive spiciness anomaly. Is there something specific about the dynamics of the region that constrain subducted layers with positive POC anomalies to have negative spiciness anomalies? Is this something that was tested? Could the fact that evidence of ESP were only found in one region/bloom be due to how subducted layers are identified? Please clarify in the text.

We thank the reviewer for these questions, and we agree that further clarification is needed here. The sign of the spiciness anomaly at depth results from both the value of spiciness at the surface (which can be both positive or negative depending on the region, see figure below) and the background value of spiciness at depth.

We ran a test where the sign of the spiciness anomaly was not imposed, and we did not find any event with positive spiciness and negative AOU anomalies. A possible explanation for this result is the latitudinal distribution of spiciness at the ocean's surface and its negative values at latitudes south of 40°S (see plot below). Omand et al. (2015) was centered at 60°N, where surface spiciness values are positive and decrease with depth. On the contrary, in the Southern Ocean surface values are negative and increase with depth between the surface and ~1000m.

Tailleux, 2021

5) I strongly advise the authors to add some material that illustrates the calculation of these different physical pumps at the same location/profile.

Figure S9 shows AOU, spice and POC in Sept 2016 but the spice anomaly is not shown (using filtered spice). Please expand on one of these figures to add the various terms used to compute the pumps and the resulting pumps: both MLP and ESP (MLD_{bio}, MLD_{dens}, AOU, mesopelagic value of POC that you remove for MLP etc.). This figure should be used to demonstrate how the calculation of ESP and MLP works and that there is no double counting.

We thank the reviewer for this suggestion. We added Figure S11 to better illustrate the calculation of the physical pumps.

6) Discuss why the eddy pump is only detected in 2016/17? How is this compatible with prior

work and your own intro putting forward the ESP as a major contributor of the global POC export.

The evaluation of the importance of the ESP relative to other mechanisms of export is a challenge that has not yet been fully solved in observational studies.

There are two sources of variability that explain the complexity of quantifying the ESP.

- 1) The first one is the small-scale nature of the water masses subducted, which means that the chances that a float captures a subduction event are low except if it is drifting in a region of high Eddy Kinetic Energy (EKE), e.g. downstream the Kerguelen Plateau (Rosso et al, 2020). The isolated nature of ESP events can explain why the eddy pump was only detected in spring 2016/17. Our float was quickly advected towards the west, hence sampling very different regions and not staying in a very high EKE region in its trajectory (see new Figure S12). However, the fact that ESP events are rare does not imply that this mechanism is not an important contributor to the BCP.
- 2) The second source of variability is the carbon flux associated with ESP events. A review of the studies that attempted to quantify this flux shows a wide range of estimates for the average contribution to the total BCP flux (Table 1). The highest estimates are the ones by Omand et al. (2015), where authors used a model to estimate that ESP contributed up to 50% of the export in spring in the high latitude oceans. The comparison with other works is difficult because Omand et al. (2015) do not provide the contribution over the annual BCP flux (as all other works do). But in any case, it shows that on average, ESP is not a major contributor (always below 30%).

Reference	Average contribution	Extreme values	Source	Time window	Domain
Omand et al, 2015	50% of spring export		Model	Spring	Southern Oc, North Atl. Kuroshio
Stukel et al, 2017	~50%		Obs/Model		California Current
Stukel et al, 2018	23%	7 to 90%	Obs/Model		California Current
Llort et al, 2018	20%	1-100%	Obs	Full-year	Southern Oc.
Boyd et al, 2019	10%	-	Model	Full-year	Global
Resplandy et al, 2019	2%		Model	Full-year	North Atl.
Davies et al, 2019	<18%	+-10%	Obs	Bloom apex	Southern Oc.
This study	19%		Obs	Bloom apex	Southern Oc.

However, what observations have also shown is that individual ESP events can be very high and able to export, in a matter of days, the amount of carbon exported by the gravitational pump during the whole year (Llort et al, 2018, Stukel et al, 2018). These estimates are only accounting for the export of POC but ESP events also export DOC. The occurrence of these “super ESP events” has

probably been miscommunicated and has contributed to the misconception that ESP is a major contributor to the BCP. We now discuss this aspect in the main text.

Figure S2b shows the strong spice anomaly signal in June 2018 but there is actually no export by the ESP at this period on Fig 3 and as stated in main text (L187: “such features were detected only during 2016-17”)! Is this because there is no POC to export at that time? Could DOC be exported in some of these spiciness events although you don’t detect POC export?

That is a useful point, and we thank the reviewer to raising it. We assume the reviewer is referring to newly-produced DOC, during the productive season, of biological origin (see Baetge et al. 2021, <https://doi.org/10.3389%2Ffmicb.2021.669883>) The role of ESP at exporting DOC, as well as surface microbial bacteria communities, is an extremely interesting topic that was already raised, yet still unresolved, by Omand et al. (2015) and Llort et al. (2018). The float we deployed for this study was not equipped with DOM sensors. It is unknown whether such sensors are sensitive enough to detect changes in biologically-produced DOC which is often a small proportion of [DOC] which is largely composed of refractory and recalcitrant DOC (Hansell and Carlson, 1998). Hence, it is non-trivial to evaluate the export of DOC. Recent works have evaluated the importance of ESP on subducting dissolved oxygen to the subsurface (Chen et al, 2021). It is then expected that future research on the ESP will focus on its role on exporting DOC. The Southern Ocean has a paucity of observations of DOC as a recent preprint supplied to the authors by Dennis Hansell attests to.

Why then do you use this event to illustrate the ESP method in Figure S2?

Do you think it is realistic that there is no subduction event at other times in such a dynamical regime as the Southern Ocean? This deserves some discussion, especially in the context of prior work including Boyd et al 2019, Omand et al 2015 and your own introduction that hypothesizes this pathway strongly contributes to carbon export. How can it be the case if for two years of sampling it is not detected?

We thank the reviewer for these questions.

The first two questions have partly been replied in Question 4’s reply. Our introduction states that ESP is “*an additional pathway*” of carbon export that contributes up to “*25% of the BGP export*”. The latter sentence is based on estimates from the literature detailed in Q4’s reply. However, it is true that the observations from our float suggest that the ESP contribution is negligible over the lifespan of the float. As detailed in Q4’s reply, we think that the low number of ESP events detected are due to the regions sampled and the trajectory of the float. We have added a sentence to tell the reader about the sensitivity of ESP observations to the method/tool employed (glider vs float).

A comprehensive investigation of the contribution of each export mechanism would require using data from as many floats as possible, homogenously covering large areas of the ocean with floats able to observe the different mechanisms. The goal of our study is to showcase how acquiring these observations is possible from a float platform, but this is not enough to provide the quantification needed to complement basin-scale estimates.

Could DOC export by these features bridge the gap of carbon export? Some discussion about this (apparent?) discrepancy is needed.

Potentially yes, but see detailed response to Question 6.

7) Discuss implications for PIPs of deep chlorophyll max in the case $MLD_{bio} > MLD_{dens}$
Please expand and discuss on the implications of the case where $MLD_{bio} > MLD_{dens}$ for your estimate of MLP and ESP.

L 358 in methods states: “.in the presence of SCM, typically in summer, MLD_{bio} can be deeper than the mixing layer depth and corresponds more likely to the base of the euphotic zone.”

a) Does that mean for MLP? IS $MLP = 0$ in this case?

b) What does this mean for the ESP? Can the eddy pump which is triggered by instabilities of MLD_{dens} or ageostrophic circulation reach down that deep if MLD_{dens} is shallower than MLD_{bio} ? Or do you actually find that ESP is close to zero in this case? If you don't find it to be zero, how do you physically explain the fact that the eddy pump would reach that deep?

Please add some material on the values of these pumps in this particular case. These are key questions for the community; your nice dataset might bring some answers even if they are partial answers. This discussion should probably be included in the main text.

The method to evaluate the MLP has totally changed. We no longer use MLD_{dens} .

8) Definition of the pseudo-Lagrangian framework

When the authors say that the float did not cross strong water mass boundaries during the period of interest, do they mean to say that each profile falls more or less along the same line in the T/S diagram? How was this determined? Can you show an example of when a float crossed a boundary? Maybe it would be easier to understand if the boundaries for the Pseudo-Lagrangian framework were added to the temperature and salinity time series (Fig. 1). Also, what does ‘station number’ (Fig. S3) mean? Is it the profile number?

We visually defined the boundaries of our three quasi-Lagrangian segments based on T/S profiles. For each of these segments, lines in the T/S diagram are much more ‘compact’ compared to all profiles together (grey lines). The best example of the float crossing a boundary is shown in Figure 1 c and d when the float moved from subpolar to polar waters on September 2016. Temperature and salinity time series with boundaries were added to Figure S3 (now Fig. S2). ‘station number’ was replaced with ‘profile number’.

9) Implications for pumps calculation during “non-Lagrangian” periods

Please clarify the implications tied to the definition of the three quasi lagrangian periods in the method section L280. The method sections states: “For completeness, the figures show the full float timeseries, but only the three quasi-Lagrangian periods are discussed in the text.”

Does this mean that we cannot trust the pumps shown in figure 3 if they are computed between these periods? If this is the case, the times outside of these three periods should be shaded or hatched or similar in all figures so the reader can identify the periods that can be interpreted robustly. Implications for these “non lagrangian periods should be discussed”

We made this clear in the method section ‘*Quasi-Lagrangian framework of the float trajectory*’

10) How can the reader get a full picture of the three pumps?

The gravitational pump is in d^{-1} while other pumps are in gC/m^2 . How can the reader compare and interpret these fluxes (Figure 3)? Some sentences that discuss this point when the authors synthesize their findings are needed. For instance, L. 223-225: ‘The contribution of all these processes over a complete annual cycle should definitely be accounted for when computing regional mesopelagic carbon budgets’. This statement is vague, and considering the detailed analysis of multiple carbon pumps in this manuscript, the authors should discuss the relative contributions of the different pump in their region, at least the two physical pumps which are given in the same unit. I realize that the gravitational pump is not provided in terms of carbon flux, but perhaps the authors can discuss rough conversion estimates or what measurements would be necessary to constrain their findings.

The first intention of this paper was to showcase the seasonal succession of the different export pathways, and the timing of each pathway with respect to phytoplankton phenology. Therefore, absolute values were not necessary. But we agree that a quantitative comparison between the different export pathways is a significant improvement for the paper. In the new version of the paper, all export pathways are now expressed in carbon-based units of $\text{mg C m}^{-2} \text{ day}^{-1}$ (see new Methods and Figure 3). We have added a paragraph in the discussion to compare the magnitude and seasonality of the fluxes over complete annual cycles.

11) Discussion, perspective and link to prior work

Some of your results lack a discussion of prior work to put them in perspective. Here are some important points to discuss:

- the timing of the MLP confined to spring restratification when biology is active and mixed layer is variable is consistent with model results from Resplandy et al 2019.

- possible interplay between PIP and BGP: you highlight that PIP might be more effective at injecting particles than the gravitational pump by passing remineralization (L210, 230). This is in apparent contradiction with Stukel et al 2017 who showed that sinking particles were more efficiently exported than smaller subducted particles that remained trapped in the upper water column. This difference in behavior might be related to the region (SO vs California Current). Note that the Dever paper you mention in your conclusion examine particle sizes that might not match the type of particles you are considering.

- the overall lack of ESP in your data after 2017 which seems in contradiction with a lot of recent works including Boyd et al Omand et al etc... (see details in comment #4).

The first and third points are now discussed in the text. Regarding the second point, we never meant to say that PIPs were more effective at injecting particles than the BGP. The main point is that the interplay between PIP and BGP can potentially boost the efficiency of the BCP. This paragraph has been reformulated.

12) Running average filter for eddy subduction pump

Fig S13 shows the depth of the eddy subduction pump. This is obtained from the filtered vertical profiles. Did the author apply a forward, backward, or forward-backward filter? We would expect

a forward or a backward filter to shift the depth of the layer. How does the phase shift affect the results shown in Fig. S13?

Backward filtering is dominated by surface and bottom MLD values and overestimating background values at depth (see figure), hence strongly reducing the magnitude of the anomalies. On the other hand, forward filtering underestimated background values along the profile and could not capture <100m features in steep vertical gradients. The centred filtering showed the most accurate isolation of anomalous features. As shown in the figure, the choice of filter type did not impact the estimated depth of the anomaly. However, it might have an impact on the magnitude of POC anomaly (see figure).

13) 159-160 ‘In all three blooms, FI was rapidly attenuated with depth (...)’ I agree with this statement for the blooms in 16/17 and 17/18, but find it debatable for 18/19. Can you speculate as to why there is a difference? The statement should be adjusted to refer to the first two blooms only.

We have no idea as to why the 18/19 Ch_l signal is less attenuated with depth. More information on the plankton community composition would be required to better understand the dynamics of particles in the mesopelagic. The statement you mention has been removed. However, we have made this clear in the figure caption (see Fig. S6).

Minor comments

L 130: “from December to February 2018”. Do you mean December 2017 or December 2018 to Feb 2019?

The SCM was observed from December 2018 to Feb 2019 but was more evident in January/February 2019. We now mention these two months in the text.

Some sentences are formulated as statements although they are probably more hypothesis. For instance L153: “Low FI and maximum bbl spike indices in mid-January 2017 following the coccolithophore bloom resulted from sinking aggregates partly made of detached coccoliths³².” Shouldn’t this be rephrased with the idea that it is compatible with the expectation that the bloom is followed by an export of detached coccoliths? You don’t actually have in-situ measurements that show these are coccoliths, do you?

Indeed, we don’t. This sentence has been removed. More generally, we have tried to be less speculative in the interpretation of our observations.

L 202: ‘All MLP events occurred prior to the bloom apex, each year (Fig. 3d ...): please label bloom stages on Fig 3 to support this statement (will also help support statement on L 218-219).

Done

L 218-219: Remind readers that ESP were only observed in a highly energetic region (but see concerns regarding negative spiciness anomaly, comments # 4)

Done

L 288: ‘Smoothened’ should be ‘smoothed’ for signal processing

Corrected

L292: I don’t think you define C_p in the text before using it in Equation 1 and where it comes from. You mention the beam attenuation later in the text (L333) but do not make a clear connection with the C_p you use for the mean particle size computation. In the float description you mention b_{bp} but not C_p . Please clarify.

This has been clarified.

Fig. 3d: The empty bars are not green on my version; please adjust the figure caption

Done

Mixed/mixing/remnant layers: I appreciate that the authors distinguish between the mixing and mixed layer. Currently, these concepts are only defined in the methods; please move the definitions earlier in the manuscript.

The mixing layer (MLD_{bio}) is now defined in the section ‘*Sampling across Southern Ocean biogeochemical provinces*’.

Diverging colormaps: The authors frequently make use of a diverging red/blue colormap for continuous variables. Unless the divergence is aligned with a meaningful threshold, this choice makes figures more difficult to interpret. Temperature, salinity, and month (Fig. 1); POC and F/b (Figs. 2, S3 & S11); and buoyancy frequency (Fig. S1) should all have continuous colormaps.

All colormaps were replaced with a continuous colormap as suggested.

Resplandy, L., Lévy, M., McGillicuddy, D.J., 2019. Effects of Eddy-Driven Subduction on Ocean Biological Carbon Pump. *Global Biogeochem. Cycles* 33, 1071–1084. <https://doi.org/10.1029/2018GB006125>

Stukel, M.R., Song, H., Goericke, R., Miller, A.J., 2017. The role of subduction and gravitational sinking in particle export, carbon sequestration, and the remineralization length scale in the California Current Ecosystem: Subduction and sinking particle export in the CCE. *Limnology and Oceanography* 63, 363–383. <https://doi.org/10.1002/lno.10636>

REVIEWERS' COMMENTS

Reviewer #1 (Remarks to the Author):

General assessment: This is my second review of this manuscript, which details a comprehensive evaluation of biogenic carbon export in the Pacific sector of the Southern Ocean from a profiling BGC-Argo float. The authors have made major improvements to the manuscript by adding a thorough description of all their methods and improving the overall quality of their plots. In this revised version, the authors made an effort to quantify carbon export (something that was missing before), providing a more direct comparison of the distinct export pathways discussed in the study. I still find somewhat problematic that all results proceed from one single float, but I also understand that not many BGC floats have all the suit of biogeochemical sensors that would allow a detailed assessment of the different export pathways. Overall, I recommend the publication of the manuscript, and have only minor comments, listed below: - Line 120: Chls and bbs are not defined before. I recommend checking that all the distinct symbols and abbreviations have been defined before presenting them. - Line 147: Could you explain why was the data missing for two months? - Line 148: "The presence of a subsurface chlorophyll maximum, characterized by high Chls / bbs values in January/February 2019 due to photoacclimation (Fig. 2b)". I honestly don't see evidence for this. The plot does not look any different in January of 2019 to any other period. - Line 262: "These relative contributions increase to 50% and 165% at the time of the bloom climax in 2016-17 and 2017-18, respectively (Fig. S13), indicating that the MLP is the dominant carbon export pathway during the pre-bloom period and strong variability in mixing layer depth." It seems that you were making the case that these PIPs were not too strong relative to gravitational settling, but now you are making the opposite statement? Or does this refer only to the pre-bloom period? I would suggest rewording this to make sure that your conclusions are expressed clearly. - Line 297. Over which depth ranges is the sampling set to each 3 m, and at what depth is the sampling resolution reduced? Also, could you please indicate if profiles are conducted at day/night or just randomly? Line 353: "The division between small and large corresponds approximately to a particle chlorophyll content of 60 pg for Chls versus Chll and a particle diameter of 100 um for bbs versus bbl.": It is not clear what sizes/concentrations do the large categories (Chll and bbl) represent. Line 359. I would recommend numbering the equations. Also, is it not problematic that the sinking rate is constant for your analysis? Is that really a justified assumption? All Figures with a time series: I recommend using letters instead of numbers as label for the Months. Figure S1. The caption refers to panel labels a), b), and c), but these are missing in the Figure. Figure S5. All the caption (not just its title) is marked as bold. Figure S9. These only-blue and only-green panels are not the best to highlight the features in the data. I recommend using a different colormap.

Reviewer #2 (Remarks to the Author):

The paper has improved in several respects. However, my main concern with this version of the manuscript is the use of a vertical sinking speed to calculate the sinking flux of particles. The sinking speeds are taken from Briggs et al.'s study for the N. Atlantic. I think there could be large deviations from these speeds, which means that the flux calculations (Fig 3) are not well substantiated. This also brings into question the paper's assessment of the different fluxes - and comparison of various pumps. I would

be a lot more comfortable if the authors left this out — or explained in the main text that the flux is based on constant sinking speed — and the results could vary by a factor of X based on the sinking speed, which is not known.

Figure 3. Using different vertical axes on the same plot for comparing fluxes is confusing. For example, Fig 2c. The axis for pulsed F is a log scale and it is difficult to compare continuous-F which is on a linear scale on the same figure. Similarly Fig 2b has one linear and another log axis.

Fig 3d. Please use the same vertical axis (scale) for both quantities.

Line 78: Should be “spike analyses” or “analysis of spikes”

Line 80: What is a “bespoke” sensor constellation?

Line 102-103: Sentence needs to be reworded - The float’s transit through different water masses makes it difficult to study seasonality of the 3 year record without consideration of spatial differences.

There is still a lack of precision and non-specificity in how terms are used in this paper.

Line 118 POC is not defined. The term is used without explaining how it is measured

Line 119 small fluorescing particles Chl_s (presumably, Chl_s is not particles, but chlorophyll-a)

Line 120 b_bs is used without any definition or explanation

Line 134-135 Diatoms are large — so should not result in Chl_s or bbs (of small particles)

Line 140: apex is not an appropriate word here

Lines 143-146 The statement is highly speculative. The authors should present evidence based results and not make statements about particle buoyancy or iron for which there are no measurements.

Line 160: There needs to be an explanation of how flux is calculated without having to refer to the Supp. Mat.

Line 207-209 and Line 56. Large fresh aggregates do not sink faster than the downward fluid velocity, so are unlikely to be transported entirely by the flow. The effect of sinking vs transport by the flow is dependent on the relative sinking speed to vertical velocity of the flow.

The vertical fluid velocity can enhance enhance the flux, depending on the slope of the particle size spectrum (Dever et al. 2019). But it is not convincing that the large aggregates are transported by the flow unless it is known that their sinking speed is small.

Reviewer #3 (Remarks to the Author):

The revised version of the manuscript is greatly improved and the authors have addressed my main concerns, specifically a better description of the methodology and more thorough assessment and discussion of the results and implications. I think this paper brings a very elegant and valuable contribution to the broad community of ocean carbon cycle and should be published.

One very minor comment:

L120: bbs is not defined in main text.

Responses to Reviewers

Manuscript: NCOMMS-21-44891A

Title: Seasonality of downward carbon export in the Pacific Southern Ocean revealed by multi-year robotic observations

Authors: Léo Lacour, Joan Lloret, Nathan Briggs, Peter G. Strutton, Philip W. Boyd

Reviewer #1:

General assessment: This is my second review of this manuscript, which details a comprehensive evaluation of biogenic carbon export in the Pacific sector of the Southern Ocean from a profiling BGC-Argo float. The authors have made major improvements to the manuscript by adding a thorough description of all their methods and improving the overall quality of their plots. In this revised version, the authors made an effort to quantify carbon export (something that was missing before), providing a more direct comparison of the distinct export pathways discussed in the study. I still find somewhat problematic that all results proceed from one single float, but I also understand that not many BGC floats have all the suit of biogeochemical sensors that would allow a detailed assessment of the different export pathways. Overall, I recommend the publication of the manuscript, and have only minor comments, listed below:

- Line 120: Chls and bbs are not defined before. I recommend checking that all the distinct symbols and abbreviations have been defined before presenting them.

We added a definition of b_{bs} . Chls was already defined line 119.

- Line 147: Could you explain why was the data missing for two months?

We suspect that the float was under seasonal sea ice and was not able to transmit data. We added this explanation in the text.

- Line 148: “The presence of a subsurface chlorophyll maximum, characterized by high Chls / bbs values in January/February 2019 due to photoacclimation (Fig. 2b)”. I honestly don’t see evidence for this. The plot does not look any different in January of 2019 to any other period.

The Chls to b_{bs} ratio is $\sim 400 \text{ mg m}^{-2}$ at the depth of MLD_{bio} (black line), and 150 mg m^{-2} at the surface during January/February 2019. It is by definition a subsurface maximum here.

- Line 262: “These relative contributions increase to 50% and 165% at the time of the bloom climax in 2016-17 and 2017-18, respectively (Fig. S13), indicating that the MLP is the dominant carbon export pathway during the pre-bloom period and strong variability in mixing layer depth.” It seems that you were making the case that these PIPs were not too strong relative to gravitational settling, but now you are making the opposite statement? Or does this refer only to the pre-bloom period? I would suggest rewording this to make sure that your conclusions are expressed clearly.

Indeed, this statement refers only to the pre-bloom period. The sentence has been reworded to make it clear.

- Line 297. Over which depth ranges is the sampling set to each 3 m, and at what depth is the sampling resolution reduced? Also, could you please indicate if profiles are conducted at day/night or just randomly?

The sampling is initially set to 5 m depth to 500 m, but it randomly varies from 3 to 10 m depending on the float ascent speed. The float profiles day and night.

Line 353: “The division between small and large corresponds approximately to a particle chlorophyll content of 60 pg for Chls versus Chll and a particle diameter of 100 μm for bbs versus bbl.”: It is not clear what sizes/concentrations do the large categories (Chll and bbl) represent.

Ch_l and b_{bl} represent particles >60 pg and >100 μm , respectively. We do not know the upper limit of particle size/concentration.

Line 359. I would recommend numbering the equations. Also, is it not problematic that the sinking rate is constant for your analysis? Is that really a justified assumption?

All Figures with a time series: I recommend using letters instead of numbers as label for the Months.

We do not refer to a specific line of an equation in the text so we think we do not need to number them. Regarding sinking speed, the assumption of a constant speed with depth is commonly made in models. The reason is that only few studies have reported depth-differentiated sinking speeds, mainly due to the difficulty in observing the settling of particles in situ.

Some studies have shown increases in mean sinking speed with depth, which would flatten the flux profile (i.e., higher at depth and lower at surface) vs. a constant sinking speed assumption. On the other hand, sinking particles tend to lose POC faster than total mass with depth, which would steepen the flux profile vs. using a single POC/bbp conversion as done in our study. Thus, the bias in flux estimates resulting from these two assumptions should cancel out.

Figure S1. The caption refers to panel labels a), b), and c), but these are missing in the Figure.

Corrected

Figure S5. All the caption (not just its title) is marked as bold.

Corrected

Figure S9. These only-blue and only-green panels are not the best to highlight the features in the data. I recommend using a different colormap.

These color bars with light colour for low values are convenient here to highlight the penetration depth of live phytoplankton cells. Features above the mixing layer depth can be seen in Figure 2.

Reviewer #2:

The paper has improved in several respects. However, my main concern with this version of the manuscript is the use of a vertical sinking speed to calculate the sinking flux of particles. The sinking speeds are taken from Briggs et al.'s study for the N. Atlantic. I think there could be large deviations from these speeds, which means that the flux calculations (Fig 3) are not well substantiated. This also brings into question the paper's assessment of the different fluxes - and comparison of various pumps. I would be a lot more comfortable if the authors left this out — or explained in the main text that the flux is based on constant sinking speed — and the results could vary by a factor of X based on the sinking speed, which is not known.

First, the sinking speed estimates from Briggs et al. are based on 18 particles export pulse events for b_{bl} and 25 events for Chl_l , including 22 events in total in the Southern Ocean. Therefore, these sinking speed estimates are representative of our study region. Second, these sinking speed estimates are used only for the calculation of gravitational flux from bio-optical spikes (Fig 3b). The second independent assessment of the gravitational flux from OST measurements (pulsed flux, Fig 3c) does not rely on the sinking speed estimates. You can see that both fluxes from the two independent methods are of a similar order of magnitude, which gives us confidence about the pertinence of the first method. Finally, the confidence interval of sinking speed estimates reported in Briggs et al. has been used in our study to evaluate the uncertainties associated with our final carbon flux estimates (see the Monte Carlo approach described in the Methods section). Error bars in Fig 3b show these uncertainties. To conclude, we present here the best flux estimates we are able to compute with present knowledge, acknowledging the limitation of the methodology.

Figure 3. Using different vertical axes on the same plot for comparing fluxes is confusing. For example, Fig 2c. The axis for pulsed F is a log scale and it is difficult to compare continuous-F which is on a linear scale on the same figure. Similarly Fig 2b has one linear and another log axis.

We have changed the vertical axis scale in Fig. 3c and 3d as suggested. However, Fig. 3b shows two different variables that we do not compare with each other. One varies by two orders of magnitude and the other one varies from 0 to 4. We have no reasons to use the same axis in this panel.

Fig 3d. Please use the same vertical axis (scale) for both quantities.

Done

Line 78: Should be “spike analyses” or “analysis of spikes”

We have changed to “analysis of spikes”

Line 80: What is a “bespoke” sensor constellation?

We mean to say that floats are versatile and can carry different type of sensors according to your mission objectives.

Line 102-103: Sentence needs to be reworded - The float's transit through different water masses makes it difficult to study seasonality of the 3 year record without consideration of spatial differences.

We would like to retain the original sentence which we believe gives the same message.

There is still a lack of precision and non-specificity in how terms are used in this paper. Line 118 POC is not defined. The term is used without explaining how it is measured

We added a definition here. But note that POC is introduced in the introduction paragraph.

Line 119 small fluorescing particles Chl_s (presumably, Chl_s is not particles, but chlorophyll-a)

We used the same term as Briggs et al. 2020 who first introduced this decomposition of the fluorescence signal. Chl_s is the fluorescence of small particles, linearly converted into chlorophyll concentration.

Line 120 b_{bs} is used without any definition or explanation

We added a definition as requested.

Line 134-135 Diatoms are large — so should not result in Chl_s or bbs (of small particles)

We disagree with the reviewer. Diatom size varies from ~2 μm (single cells) to few mm (long chains). Thus, diatoms can potentially be detected in the fraction of small (<100 μm) backscattering and fluorescing particles (b_{bs} and Chl_s).

Line 140: apex is not an appropriate word here

We don't understand why 'apex' would not be appropriate here. In our paper, we define the apex as the annual maximum phytoplankton carbon stock. This maximum occurs in December 2017.

Lines 143-146 The statement is highly speculative. The authors should present evidence based results and not make statements about particle buoyancy or iron for which there are no measurements.

This statement gives an example on how the iron status can influence particle characteristics. It is based on observations from Boyd et al. 2005, not from our results. We slightly modified the sentence to make this clear.

Line 160: There needs to be an explanation of how flux is calculated without having to refer to the Supp. Mat.

The methods are first introduced in the introduction paragraph, described again in figure captions and explained in details in the Methods section (not in the supplementary material). We think adding the methods in the result paragraph would break the flow of the paper.

Line 207-209 and Line 56. Large fresh aggregates do not sink faster than the downward fluid velocity, so are unlikely to be transported entirely by the flow. The effect of sinking vs transport by the flow is dependent on the relative sinking speed to vertical velocity of the flow. The vertical fluid velocity can enhance the flux, depending on the slope of the particle size spectrum (Dever et al. 2019). But it is not convincing that the large aggregates are transported by the flow unless it is known that their sinking speed is small.

Based on observations from Briggs et al. 2020, typical sinking speeds for large aggregates, identified as optical spikes, range from 58 to 129 m d⁻¹. In a recent study (paper in preparation), we did observe aggregates, from 100 to 2500 μm, settling at a maximum speed of 150 m d⁻¹ in the Southern Ocean using UVP (Underwater video profiler) images. Therefore, typical downwelling velocities on the order of 100 m d⁻¹ generated at submesoscales (Fox-Kemper et al. 2008, Mahadevan 2016) are similar to large aggregate sinking velocities. As mentioned in Dever et al. 2021, “when the gravitational sinking velocity of particles is comparable to (or smaller than) the vertical velocities in the flow field, the dynamics of the flow field can impact the trajectories and fate of the POC”. Based on these results, we believe that the hypothesis that ESP events can transport large aggregates to depth is valid.

Reviewer #3:

The revised version of the manuscript is greatly improved and the authors have addressed my main concerns, specifically a better description of the methodology and more thorough assessment and discussion of the results and implications. I think this paper brings a very elegant and valuable contribution to the broad community of ocean carbon cycle and should be published.

One very minor comment:

L120: bbs is not defined in main text.

We added a definition in the main text.